# Three-dimensional CRISPR screening reveals epigenetic interaction with anti-angiogenic therapy

Michael Y. He [1,2,10], Michael M. Halford[1], Ruofei Liu[1,2], James P. Roy[1,2], Zoe L. Grant[3,4,11], Leigh Coultas [3,4], Niko Thio[5], Omer Gilan [2,6,12], Yih-Chih Chan [6], Mark A. Dawson [2,6,7,8], Marc G. Achen[1,2,9,13] & Steven A. Stacker [1,2,9✉]

Angiogenesis underlies development, physiology and pathogenesis of cancer, eye and cardiovascular diseases. Inhibiting aberrant angiogenesis using anti-angiogenic therapy (AAT) has been successful in the clinical treatment of cancer and eye diseases. However, resistance to AAT inevitably occurs and its molecular basis remains poorly understood. Here, we uncover molecular modifiers of the blood endothelial cell (EC) response to a widely used AAT bevacizumab by performing a pooled genetic screen using three-dimensional microcarrier-based cell culture and CRISPR–Cas9. Functional inhibition of the epigenetic reader BET family of proteins BRD2/3/4 shows unexpected mitigating effects on EC survival and/or proliferation upon VEGFA blockade. Moreover, transcriptomic and pathway analyses reveal an interaction between epigenetic regulation and anti-angiogenesis, which may affect chromosomal structure and activity in ECs via the cell cycle regulator CDC25B phosphatase. Collectively, our findings provide insight into epigenetic regulation of the EC response to VEGFA blockade and may facilitate development of quality biomarkers and strategies for overcoming resistance to AAT.

[1] Tumour Angiogenesis and Microenvironment Program, Peter MacCallum Cancer Centre, Melbourne, VIC, Australia. [2] Sir Peter MacCallum Department of Oncology, The University of Melbourne, Parkville, VIC, Australia. [3] Epigenetics and Development Division, The Walter and Eliza Hall Institute of Medical Research, Parkville, VIC, Australia. [4] Department of Medical Biology, The University of Melbourne, Parkville, VIC, Australia. [5] Bioinformatics Core, Peter MacCallum Cancer Centre, Melbourne, VIC, Australia. [6] Translational Haematology Program, Peter MacCallum Cancer Centre, Melbourne, VIC, Australia. [7] Centre for Cancer Research, The University of Melbourne, Parkville, VIC, Australia. [8] Department of Haematology, Peter MacCallum Cancer Centre, Melbourne, VIC, Australia. [9] Department of Surgery, Royal Melbourne Hospital, The University of Melbourne, Parkville, VIC, Australia. [10] Present address: Princess Margaret Cancer Centre, University Health Network, Toronto, ON, Canada. [11] Present address: Gladstone Institutes, San Francisco, CA, USA. [12] Present address: Australian Centre for Blood Diseases, Monash University, Melbourne, VIC, Australia. [13] Present address: St Vincent's Institute of Medical Research, Melbourne, VIC, Australia. ✉email: Steven.Stacker@petermac.org

Angiogenesis represents the formation of new blood vessels from pre-existing vasculature and plays vital roles in development, physiology, and pathophysiology[1]. Inhibiting aberrant angiogenesis using anti-angiogenic therapy (AAT) has demonstrated efficacy in the treatment of many human diseases[2,3]. For example, bevacizumab, the neutralizing monoclonal antibody (mAb) against vascular endothelial growth factor A (VEGFA), has been approved for treating seven different advanced human cancers (with or without chemotherapy) and is widely used off-label in the treatment of various ocular diseases such as age-related macular degeneration as a common alternative to the approved anti-VEGFA antibody ranibizumab[2,4]. Furthermore, recent evidence indicates a role for bevacizumab to potentiate current immunotherapy (e.g., the anti-PD-L1 antibody atezolizumab) in human patients with metastatic solid tumors[5,6]. However, clinical benefit from AAT treatment has been limited largely by variable and unpredictable responses as well as the inevitable occurrence of resistance[3,4].

Understanding patient response to AAT treatment and the mechanisms of resistance has proved challenging as many cell types in the body (e.g., endothelial, immune, stromal, and cancer cells) are responsive to, for example, VEGFA signaling[7,8]. This is further complicated by the highly dynamic interactions between these cell types and the heterogeneity seen between patients as well as within individual patient's tumors[9]. Microvascular blood endothelial cells (ECs) are at the core of angiogenesis and play an important role in promoting disease progression and shaping therapeutic responses[8,10–12]. Hence, deciphering the role of ECs in patient response to AAT could provide insight into how individual patients respond differently and how resistance occurs and/or develops.

In vitro pooled genetic screening using CRISPR–Cas9 is a powerful approach for interrogation of gene function in biological systems[13]. Specifically, it can be used to identify genes responsible for drug resistance and/or facilitate the discovery of essential genes that can be targeted to improve therapeutic effects in a high-throughput, systematic, and unbiased manner[14]. A discovery strategy using a pooled genetic screening approach is therefore an appropriate tool to uncover the role of ECs and their response modifiers to AAT. However, there is a current vacancy with respect to pooled genetic screening in human microvascular blood ECs partly because in vitro cultivation of ECs is difficult at a large scale[15].

Epigenetic regulation of angiogenesis has been increasingly appreciated over recent years and this includes modulation of histone and/or transcription factor modification and DNA methylation in genomic regions important for transcription initiation or in intragenic regions[16–18]. Of those, the bromodomain and extra terminal domain (BET) family of proteins (including BRD2, BRD3, BRD4, and BRDT in humans), which acts as an epigenetic reader to recognize and bind to acetyl-lysine residues on histones and transcription factors, has been shown to regulate gene transcription and many cellular activities including angiogenesis[18–20]. Although emerging data show that many small-molecule inhibitors targeting epigenetic signaling have an anti-angiogenic property and resistance to AAT potentially has a reversible (epigenetic) nature[3,19,21,22], understanding of the interaction between epigenetics and anti-angiogenesis needs to be further developed.

In this study, we identify and validate molecular modifiers of the EC response to bevacizumab using a systematic and minimally biased screening approach. We adopt a three-dimensional (3D) microcarrier-based culture to address the high demand for cell numbers in the pooled genetic screen. We then confirm the involvement of epigenetic regulation in modifying the EC response to bevacizumab by unveiling a previously unreported interaction between BET protein activity and VEGFA signaling. Importantly, these observations will prompt further investigation into the role of epigenetic regulation in vascular biology, pathological angiogenesis, and response to AAT. Our findings could facilitate the clinical development of predictive and/or response biomarkers and strategies for overcoming therapeutic resistance, ultimately enabling the rational use of AAT. A flow chart summarizing the biological models and technical recourses, experimental outputs, as well as research impact of this study is included in Supplementary Fig. 1.

## Results

**3D microcarrier-based culture of human microvascular blood ECs.** Angiogenesis usually occurs in the microvasculature such as capillaries that are mainly composed of the slow-growing and phenotypically faithful microvascular blood ECs[12]. This type of EC is therefore highly relevant to physiology but difficult to culture in vitro. We, therefore, adopted a microvascular blood EC line immortalized with human telomerase reverse transcriptase in our study (see Methods for further details of this cell line). The current model for in vitro cultivation of ECs is based on a 2D monolayer culture system where cells attach to a flat surface (usually with a small surface-area-to-volume ratio, 3.3–5 cm²/mL) and grow in a static environment (Fig. 1a, b). Generating a large number of viable ECs is therefore challenging using the 2D model. To meet the high demand for cell numbers in a pooled genetic screen, we developed a 3D microcarrier-based culture system. This incorporated a large surface-area-to-volume ratio (at least 14.4 cm²/mL in our system) and flow-induced shear stress so that cells can attach, grow and spread on microcarriers and actively migrate and form bridges between microcarriers (Fig. 1a, b; Supplementary Fig. 2a). We additionally demonstrated that EC proliferation and baseline gene expression in complete growth medium were comparable between 2D and 3D cultures (Fig. 1c; Supplementary Fig. 2b and Supplementary Data 1). Taken together, we confirmed that our 3D microcarrier-based culture system was capable of promoting proliferation of microvascular blood ECs as well as simulating in vivo conditions such as shear stress and EC migration, while without altering their baseline gene expression. This established a biological foundation for identifying molecular modifiers of the EC response to AAT.

**Identification of EC response modifiers to bevacizumab.** CRISPR–Cas9 is a powerful and versatile technology for manipulating gene function in biological systems[23]. We, therefore, evaluated CRISPR–Cas9 in our microvascular blood ECs (Supplementary Figs. 3 and 7) and the quality of Cas9/single guide RNA (sgRNA) library in the plasmid pool (Supplementary Fig. 4a). To study the EC response to VEGFA blockade, we combined these elements with microcarrier-based culture to perform a 3D kinome-wide (targeting genes encoding both typical and atypical kinases) CRISPR screen following the workflow shown in Fig. 2a. An important feature of our screen was VEGFA-dependent culture conditions achieved by using endothelial serum-free medium (ESFM), which enabled phenotypic selection to be specific to bevacizumab treatment. Screening conditions were evaluated at the key steps during the screen. Deep sequencing of PCR amplicons prepared using genomic DNA (gDNA) isolated on day 0 confirmed effective transduction for sgRNA integration into the host EC genome with representation maintained to that in the plasmid pool (Supplementary Fig. 4b). During the steps before screening selection (day −9 to day 0, Fig. 2b), EC proliferation was in agreement with the previous results of microcarrier-based culture (Fig. 1c) with a slight decrease in viable cell number during antibiotic selection and

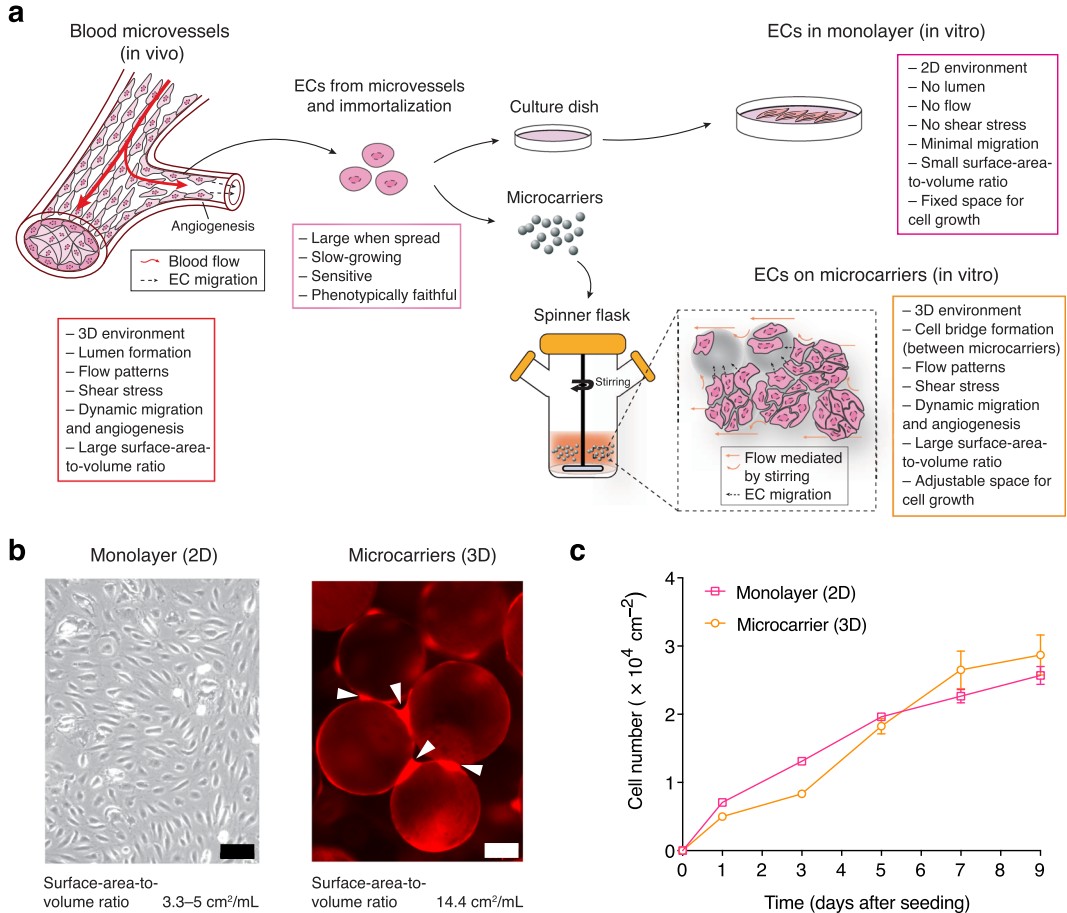

**Fig. 1 In vitro culture of immortalized human microvascular blood ECs. a** Schematics of strategies for in vitro culture of human microvascular blood ECs. ECs are isolated from blood vessels and immortalized for in vitro culture where they are cultivated in monolayer using flat-surface culture vessels or on microcarriers stirred in spinner flasks. **b** Representative images of ECs cultivated using monolayer culture (parental ECs) or microcarrier-based culture (parental ECs expressing a red fluorescent protein DsRed-Max). White arrowheads indicate cell bridges that suggest bead-to-bead transfer. Scale bar (black), 100 μm. Scale bar (white), 50 μm. **c** Cell number assessed from monolayer culture and microcarrier-based culture of ECs. Error bars represent ± SEM ($n = 3$ independent experiments).

serum reduction. During screening selection (day 0–21, Fig. 2c), differential EC survival was observed between bevacizumab and palivizumab (the negative isotype-matched antibody control) treatment, confirming that VEGFA-dependent culture conditions were maintained during this period.

Next, we processed the isolated gDNA by deep sequencing for analysis of differential sgRNA representation. A modified robust rank aggregation (α-RRA) score was calculated according to the *P* value-based sgRNA ranking and corrected for multiple hypothesis testing to identify candidate genes[24]. If a gene is associated with a small α-RRA score, it indicates that the sgRNAs targeting this gene are ranked consistently high and this gene is likely a candidate for the resultant phenotype. In addition, if the sgRNAs are enriched or depleted, the gene targeted would be defined to be identified from positive or negative selection, respectively. At a false discovery rate (FDR) ≤ 0.3, a total of 18 candidate genes with small α-RRA scores were identified from positive or negative selection (as potential resistance-mediators or sensitizers of the EC response to bevacizumab, respectively) at day 12 (Fig. 2d) or day 21 (Fig. 2e). The detailed information of the candidate genes and screening analysis is included in Supplementary Data 2 and 3, respectively.

**Validation of EC response modifiers to bevacizumab.** For the first step of validation, we assessed whether the candidate genes

were expressed in ECs. RNA-Seq analysis confirmed that all candidate genes were highly expressed at the mRNA level (compared with the housekeeping gene *GAPDH*) except *CDKL3*, whose expression was relatively low (Fig. 3a and Supplementary Data 1). In addition, given that *BRD4* and *BRDT* are closely related to *BRD2* and *BRD3*, their expression was also examined. The expression of *BRD4* was detected at a similar level to that of *BRD2* and *BRD3* whereas *BRDT* was not expressed in ECs (Fig. 3a). To perform functional validation, we examined siRNA-mediated knockdown of all candidate genes plus *BRD4* in the endothelial cell-multicolor competition assay (EC-MCA), which served as a complementary system to the CRISPR screen to assess the EC response to bevacizumab (Fig. 3b). The conditions for siRNA-mediated gene knockdown in ECs were initially evaluated using a siRNA pool targeting *GAPDH* (siGAPDH) (Supplementary Figs. 5a, b and 8) and later validated using siBRD2, siBRD3, and siBRD4 by quantitative reverse transcription PCR to confirm a knockdown efficiency of ~50%, 80%, and 90%, respectively (Supplementary Fig. 5c). In the EC-MCA, a final ratio (*M*) was generated from flow cytometry data analyzed using the gating strategy displayed in Fig. 3c. Among all targets examined, *BRD2*, *BRD3*, *BRD4*, *TAOK1*, *ACTR2,* and *TRRAP* knockdown generated *M* that were significantly > 1 (mediating EC resistance to bevacizumab), whereas *TLK1* and *TLK2* knockdown generated *M* that were significantly < 1 (sensitizing ECs to bevacizumab)

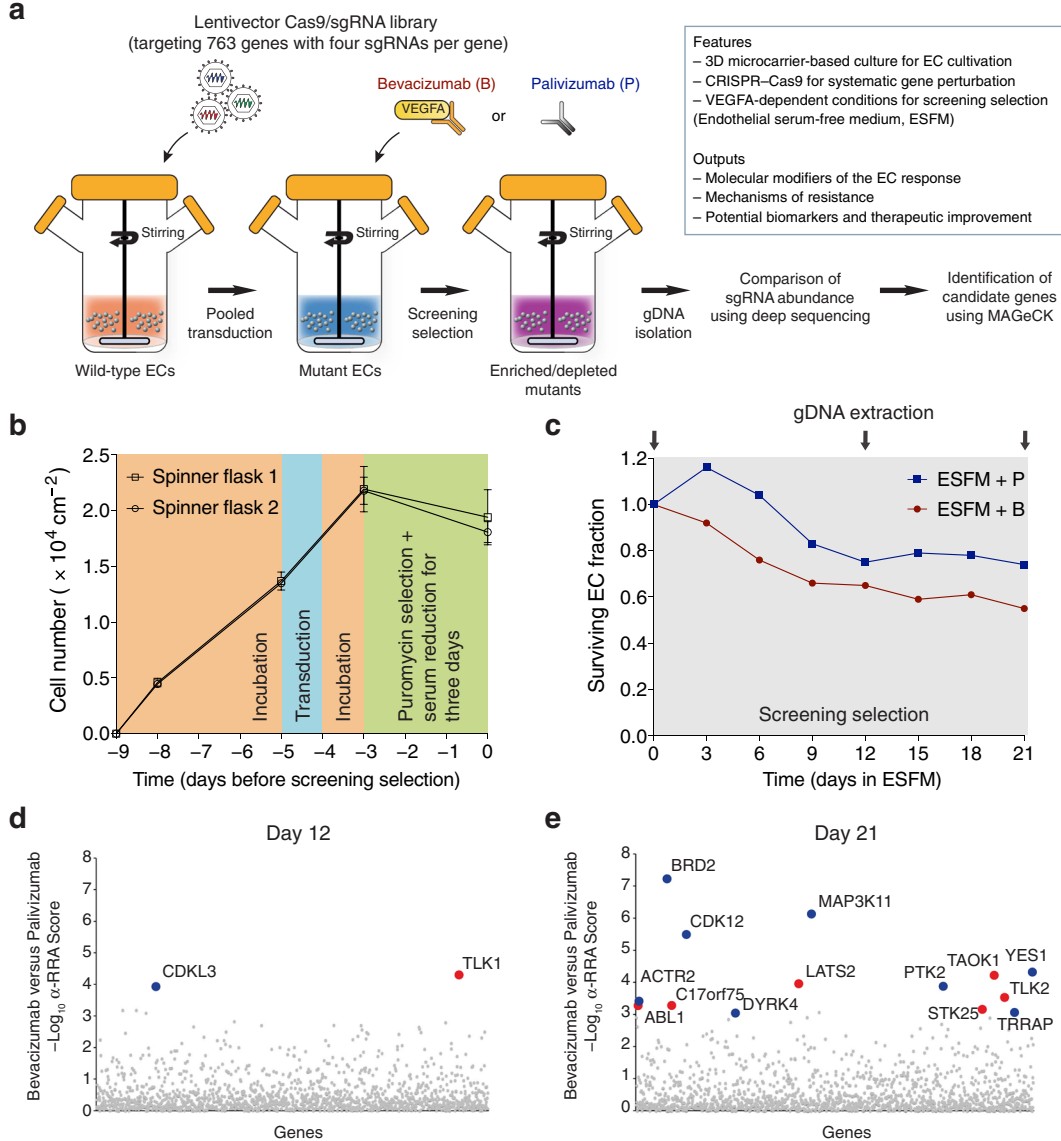

**Fig. 2 Identification of EC response modifiers to bevacizumab. a** Workflow of the 3D kinome-wide CRISPR screen. **b** Viable cell number determined before screening selection. Error bars represent ± SEM ($n = 2$ independent experiments). **c** Cell survival during screening selection. Surviving EC fraction is the ratio of relative viable cell number on a specific day versus that on day 0. Curves represent results from one experiment. **d**, **e** Identification of candidate genes. Candidate genes were identified using the α-RRA score (shown as −log₁₀ α-RRA score) on day 12 (**d**) or 21 (**e**). Blue or red dots represent candidate genes (FDR ≤ 0.3) identified from positive or negative selection (bevacizumab versus palivizumab), respectively. Small gray dots represent the neutral genes. Results represent two independent experiments ($n = 2$).

(Fig. 3d). These results were consistent with the screen data except for *TAOK1*, whose loss-of-function (LOF) was shown to induce EC sensitization to bevacizumab in the screen.

To further validate the results of the screen and the EC-MCA as well as to facilitate candidate selection for characterization, we evaluated small-molecule inhibitors including the pan-BET bromodomain inhibitors (BETi) JQ1 and I-BET762; the TAOK kinase inhibitor compound 43; and the tousled-like kinase 1 (TLK1) kinase inhibitor thioridazine (THD) in the small-molecule inhibitor assay (SMIA). Similar to the EC-MCA, the SMIA generated a final ratio (S) to indicate the effect of small-molecule inhibitor treatment on the EC response to bevacizumab (Fig. 3e). In agreement with the EC-MCA results, JQ1 (at all concentrations tested) and I-BET762 (at 1000 nM) phenocopied siRNA-mediated knockdown of individual BET protein but with a more profound effect (S > 1.5) (Fig. 3f). Compound 43 (at 5, 10, and 20 µM) also confirmed the effect of *TAOK1* knockdown (S > 1) (Fig. 3f). In

contrast, THD (at all concentrations tested) conferred EC resistance to bevacizumab (S > 1) (Fig. 3f), which was opposite to the results of the screen and the EC-MCA.

Overall, our results from two independent experimental systems (pooled genetic screen versus arrayed cell assay) using three different methods for functional perturbation (CRISPR–Cas9-mediated gene knockout versus siRNA-mediated gene knockdown versus pharmacological inhibition) consistently showed that *BRD2/3/4* LOF or inhibition of BET protein activity conferred EC resistance to bevacizumab. We then prioritized BET proteins for characterization.

**BET inhibition suppresses EC activities.** We next explored the role of BET proteins in ECs under the normal growth conditions (using the endothelial cell growth medium-2 microvascular (EGM-2MV) medium). We first evaluated the effects of siRNA-mediated

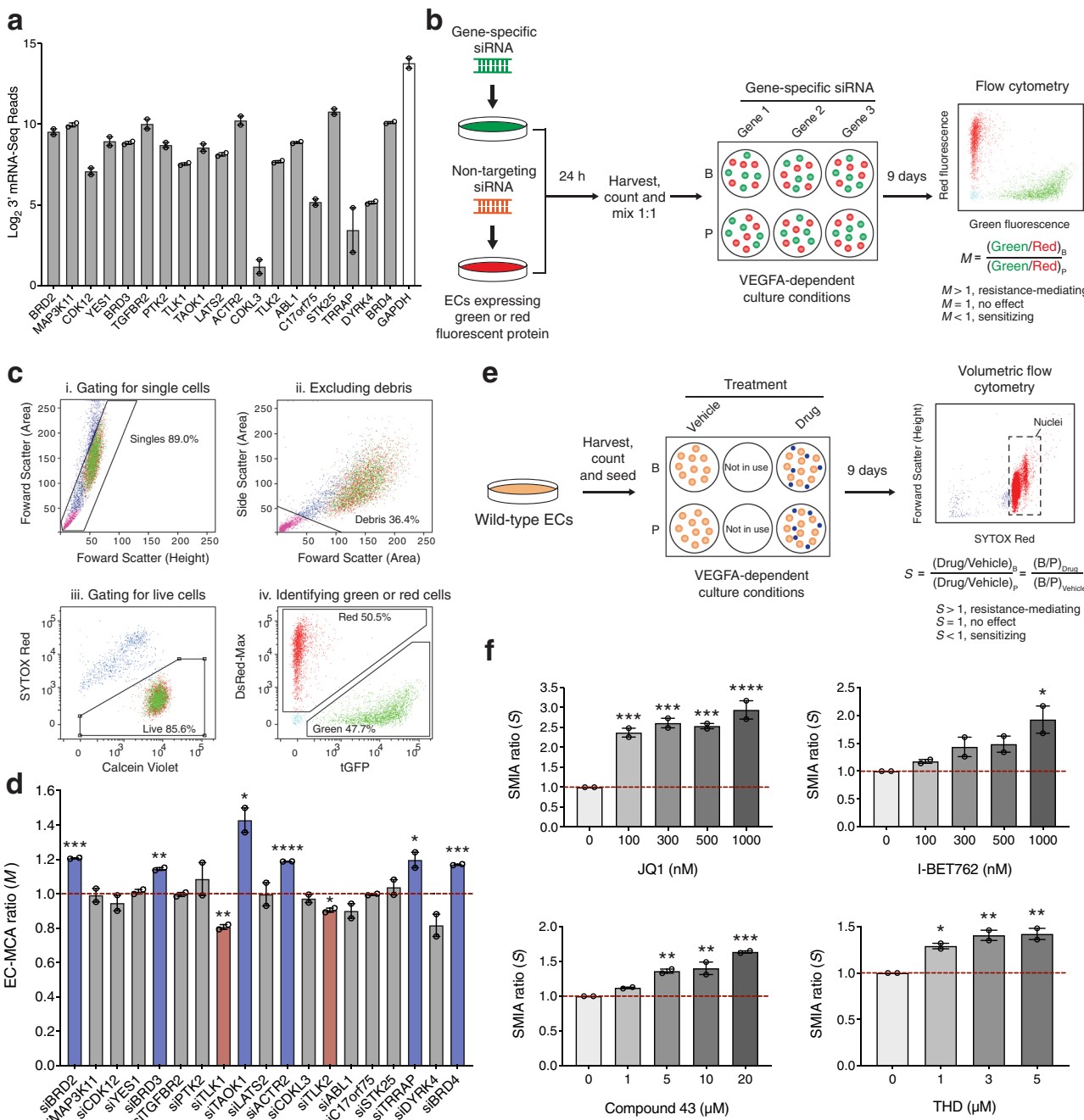

**Fig. 3 Validation of screen candidate genes. a** Baseline expression of the candidate genes and *BRD4* was assessed in ECs cultivated in microcarrier-based culture using RNA-Seq. Gene expression is represented by $\log_2$ 3′ mRNA-Seq reads. Expression of *GAPDH* is used as a reference for relative expression (white bar). All genes were highly expressed except *CDKL3*. Error bars represent ± SEM ($n = 2$ independent experiments). **b** Schematic representation of the EC-MCA (see Methods for further details). **c** Gating strategy of flow cytometry-based determination of *M*. Total number of events per sample was 10,000. The percentages shown are those of the parent population. **d** EC-MCA analysis of the effect of siRNA-mediated knockdown of the candidate genes and *BRD4*. A ratio of 1 (red dotted line) indicates no effect of siRNA on the EC response to bevacizumab. $M > 1$, resistance-mediating effect. $M < 1$, sensitization. Blue or red bars represent the resistance-mediating or sensitizing candidates identified from the EC-MCA, respectively. Error bars represent ± SEM ($n = 2$ independent experiments). *$P$ value < 0.05; **$P$ value < 0.01; ***$P$ value < 0.001; ****$P$ value < 0.0001 (versus $M = 1$, two-tailed unpaired $t$ test). **e** Schematic representation of the SMIA (see Methods for further details). **f** Effects of pan-BETi, JQ1, and I-BET762; TAOK kinase inhibitor, compound 43; and TLK1 kinase inhibitor, THD were assessed using the SMIA. A ratio of 1 (red dotted line) indicates no effect on the EC response to treatment. $S > 1$, resistance-mediating effect. $S < 1$, sensitization. Error bars represent ± SEM ($n = 2$ independent experiments). *, adjusted $P$ value < 0.05; **, adjusted $P$ value < 0.01; ***, adjusted $P$ value < 0.001; ****, adjusted $P$ value < 0.0001 (versus $S = 1$, one-way ANOVA with Dunnett's multiple comparison test performed).

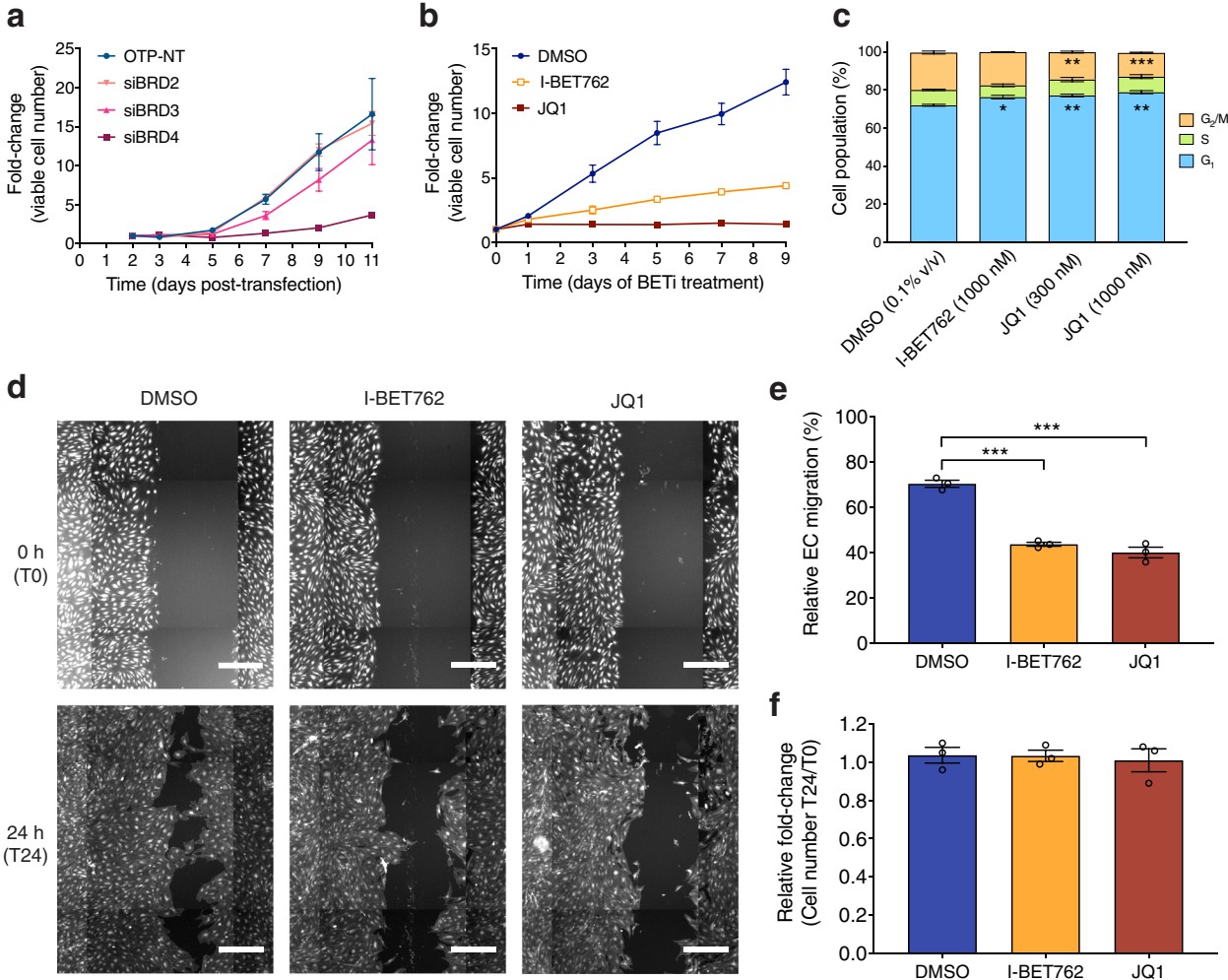

**Fig. 4 Targeting BET protein suppresses key EC activities under normal growth conditions. a** Effects of siRNA-mediated knockdown of *BRD2*, *BRD3*, or *BRD4* on survival and/or proliferation of ECs. Error bars represent ± SEM (*n* = 2 independent experiments). **b** Effects of I-BET762 (1000 nM) or JQ1 (300 nM) on survival and/or proliferation of ECs. Error bars represent ± SEM (*n* = 2 independent experiments for all conditions except *n* = 4 independent experiments for DMSO). **c** Effects of BETi on nuclear DNA content of ECs. Nuclei were released and analyzed on day 3 of BETi treatment. Error bars represent ± SEM (*n* = 2 independent experiments for all conditions except *n* = 3 independent experiments for DMSO). *, adjusted *P* value < 0.05; **, adjusted *P* value < 0.01; ***, adjusted *P* value < 0.001 (versus DMSO, one-way ANOVA with Dunnett's multiple comparison test performed). **d** Scratch wound migration assay analysis of the effect of I-BET762 (1000 nM) or JQ1 (300 nM) on EC migration. For imaging, ECs were stained using CellTracker Green CMFDA dye at 0 h (T0) then fixed and stained with phalloidin CF488A and Hoechst 33342 at 24 h after scratching (T24). Images were assembled as collages. Scale bars, 200 μm. **e** Quantification of EC migration. **f** Quantification of cell number at T24 normalized to T0. Error bars represent ± SEM (*n* = 3 independent experiments). ****P* value < 0.001 (two-tailed unpaired *t* test).

knockdown and BETi on important EC activities. Both siBRD4 (but not siBRD2 or siBRD3) and BETi (I-BET762 at 1000 nM or JQ1 at 300 nM) resulted in lower viable cell numbers compared with negative control over nine days of incubation (Fig. 4a, b), suggesting a survival- and/or proliferation-inhibiting effect. Consistent with their effects on EC survival and/or proliferation, BETi disrupted cell cycle progression in ECs. After three days of treatment, I-BET762 at 1000 nM and JQ1 at both low and high concentrations (300 and 1000 nM) significantly enriched the $G_1$ cell population, and both concentrations of JQ1 further significantly reduced the $G_2/M$ cell population compared with vehicle treatment (dimethyl sulfoxide, DMSO) (Fig. 4c). In addition, we evaluated the effect of BETi on EC migration using a scratch wound migration assay. Twenty-four hours after scratching, wound closure in DMSO was ~70% of the initial wound area whereas both I-BET762 at 1000 nM and JQ1 at 300 nM reduced the closed area to ~40% (Fig. 4d, e). These results are mainly attributable to EC migration because EC survival and/or proliferation were not

affected within 24 h after scratching (Fig. 4f). BETi therefore significantly suppressed EC migration. Collectively, our data show that targeting BET proteins inhibits vital activities of microvascular blood ECs.

**Interaction between BET inhibition and VEGFA blockade causes complex mitigating effects in ECs.** Having demonstrated that targeting BET proteins using BETi conferred EC resistance to VEGFA blockade but had an anti-angiogenic effect, we next sought to evaluate whether there was an interaction between BET inhibition and VEGFA blockade. To accomplish this, we assessed different treatment combinations in a range of culture conditions using the SMIA format (Fig. 3e) across multiple time points. These included the EC complete growth medium EGM-2MV (containing high serum concentration and extra growth factors), the serum-free medium ESFM (containing no serum but extra growth factors, serum supplements, and cancer cell-conditioned medium), and

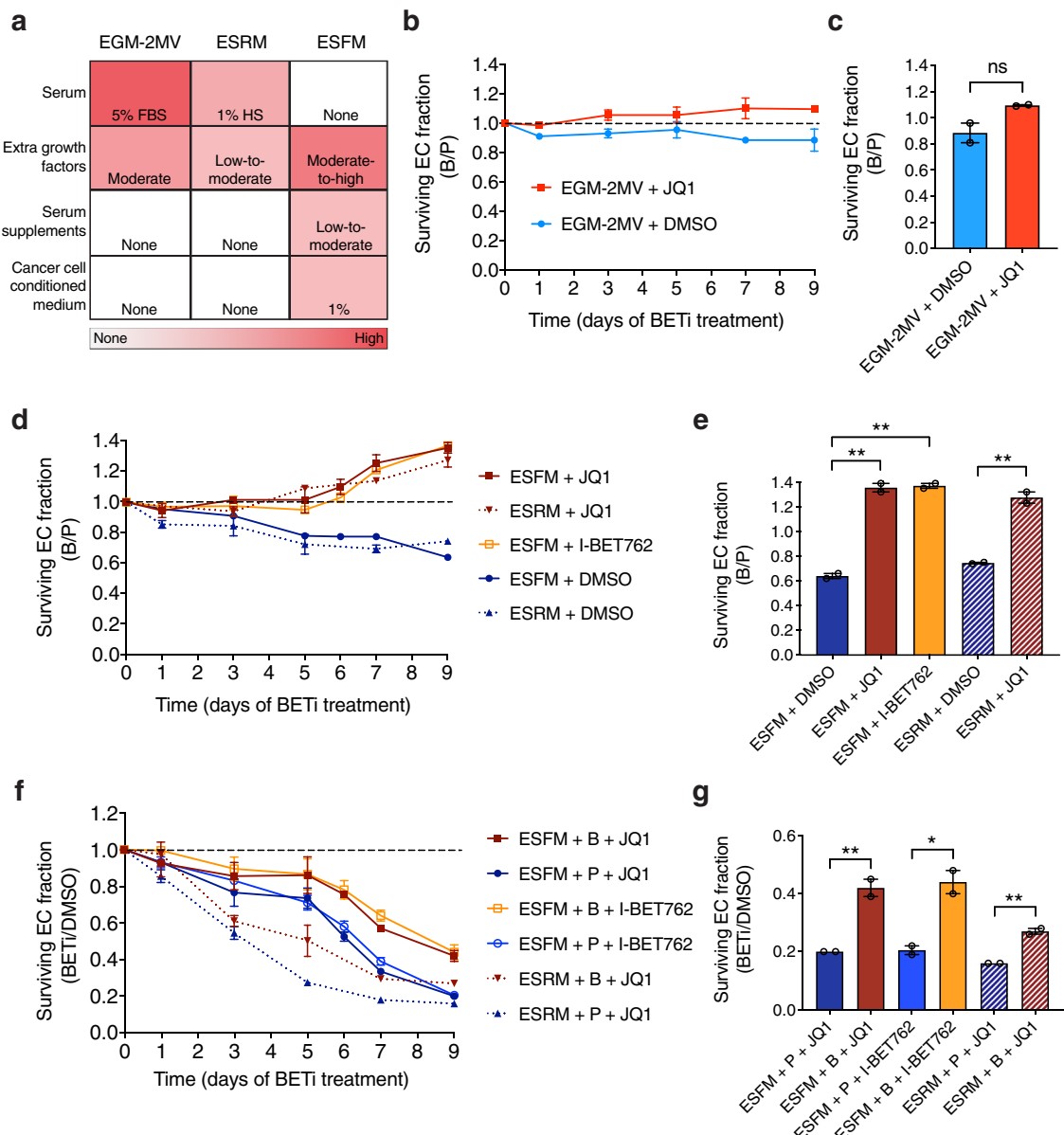

**Fig. 5 Mitigating interaction between BETi and bevacizumab in ECs. a** Characteristics of VEGFA-independent (EGM-2MV) and VEGFA-dependent (ESRM and ESFM) culture conditions. FBS, fetal bovine serum; HS, human serum. **b, c** Relative cell response to bevacizumab in EGM-2MV plus DMSO or JQ1 during days 0–9 (**b**) or on day 9 (**c**): JQ1 did not change EC response to bevacizumab under VEGFA-independent conditions. **d, e** Relative cell response to bevacizumab in ESFM or ESRM plus DMSO or BETi during days 0–9 (**d**) or on day 9 (**e**): BETi reversed EC sensitivity to bevacizumab under VEGFA-dependent conditions. **f, g** Relative cell response to BETi in ESFM or ESRM plus palivizumab or bevacizumab during days 0–9 (**f**) or on day 9 (**g**): bevacizumab attenuated survival- and/or proliferation-inhibiting effect(s) of BETi under VEGFA-dependent conditions. A ratio of 1 (black dotted line) indicates no effect on survival and/or proliferation of ECs. Error bars represent ± SEM ($n = 2$ independent experiments). *$P$ value < 0.05; **$P$ value < 0.01; ***$P$ value < 0.001; ****$P$ value < 0.0001; ns, not significant (two-tailed unpaired $t$ test).

endothelial serum-reduced medium (ESRM) (containing low serum concentration and extra growth factors) (Fig. 5a). The relative EC response to bevacizumab was expressed as a ratio calculated as surviving EC fraction in medium plus bevacizumab versus that in medium plus palivizumab (referred to as the B/P ratio). In EGM-2MV, the difference between the B/P ratio in DMSO on day 9 and the ratio 1 (i.e., no effect) was not significant ($P$ value = 0.28; two-tailed unpaired $t$ test) (Fig. 5b), indicating that ECs were insensitive to bevacizumab in EGM-2MV and the culture conditions were therefore VEGFA-independent. In addition, the B/P ratio under JQ1 treatment did not differ from that in DMSO on day 9 ($P$ value = 0.13; two-tailed unpaired $t$ test) (Fig. 5c),

demonstrating that JQ1 did not change the EC response to bevacizumab under VEGF-independent culture conditions. However, under the previously defined VEGFA-dependent culture conditions (ESFM), the B/P ratio under either I-BET762 or JQ1 treatment constantly increased and was significantly higher than that in DMSO on day 9 (Fig. 5d, e), suggesting that BETi altered the EC response by mitigating sensitivity to bevacizumab in ESFM. To exclude the possibility that our results were due to artifacts of the serum-free culture conditions, we additionally evaluated all treatment combinations in alternative VEGFA-dependent culture conditions (ESRM). Consistently, the B/P ratios in ESRM plus DMSO or JQ1 reproduced those in ESFM at all time points tested

(Fig. 5d, e) despite the very different compositions of these two media (Supplementary Tables 3 and 4). These results indicate that the effects of BETi on the EC response to bevacizumab were not restricted to specific conditions and can be detected in different VEGFA-dependent culture conditions but not in VEGFA-independent culture conditions.

To further evaluate whether bevacizumab had an effect on the EC response to BETi, we reanalyzed the data in Fig. 5d from a BETi perspective. The relative cell response to BETi was expressed as a ratio calculated as surviving EC fraction in medium plus BETi versus that in medium plus DMSO (referred to as the BETi/DMSO ratio). All BETi/DMSO ratios regardless of mAb treatment and medium composition continuously decreased such that all were below 0.5 on day 9 (Fig. 5f), indicating that ECs were highly sensitive to BETi treatment. However, the BETi/DMSO ratio in medium plus bevacizumab was significantly higher than that in medium plus palivizumab on day 9 (Fig. 5g), suggesting a desensitizing effect of bevacizumab on the EC response to BETi.

Taken together, our results suggest that BET inhibition and VEGFA blockade mitigate the survival- and/or proliferation-inhibiting effects caused by each other in ECs via an interacting mechanism. ECs co-treated with BETi and bevacizumab obtained an advantage for survival and/or proliferation compared with those treated with BETi and palivizumab. Intriguingly, this is consistent with maintaining a normal cell morphology whether under co-treatment with BETi and bevacizumab or in DMSO, and these cells did not exhibit the heterogeneous shape changes which may be caused by primary (e.g., through changing intracellular signaling) and/or secondary effects (e.g., through inducing low cell density and/or limiting paracrine signaling) of BET inhibition alone (Supplementary Fig. 6).

**Gene expression alterations in ECs co-treated with BETi and bevacizumab**. To explore potential mechanisms underlying the altered EC response caused by co-treatment with BETi and bevacizumab, we performed an RNA-Seq analysis of ECs treated with DMSO or BETi (I-BET762 at 1000 nM or JQ1 at 300 nM) in ESFM plus bevacizumab or palivizumab (Fig. 6a). An initial list of differentially expressed genes (DEGs) between bevacizumab plus I-BET762 or JQ1 (referred to as B_I-BET or B_JQ1, respectively) versus palivizumab plus I-BET762 or JQ1 (referred to as P_I-BET or P_JQ1, respectively) was generated from the RNA-Seq analysis of three independent experiments using the criteria: FDR < 0.05 and |$\log_2$ fold-change| > 1. To ensure DEGs were specific to the comparison between bevacizumab plus BETi versus palivizumab plus BETi, we further excluded those that appeared in the comparison between bevacizumab plus DMSO (referred to as B_DMSO) versus palivizumab plus DMSO (referred to as P_DMSO) to obtain the final list of DEGs. The full DEG analysis across all different comparisons is included in Supplementary Data 4. A heat map of relative expression of the final DEGs was generated using mean-centering and unsupervised hierarchical clustering (Fig. 6b). The heat map shows that samples from cells treated with palivizumab plus BETi clustered independently from those treated with mAb plus DMSO and bevacizumab plus BETi (Fig. 6b), indicating that the transcriptomic profile of cells treated with bevacizumab plus BETi resembled that of cells in DMSO. In addition, all the final DEGs were highlighted in the volcano plots and CDC25B (encoding the cell cycle regulator CDC25B phosphatase) was identified as the most significantly upregulated gene in both B_I-BET and B_JQ1 (Fig. 6c).

RNA-Seq data were further analyzed for evaluation of gene set enrichment. All RNA-Seq data were processed and correlated with a priori defined gene sets which were compared between bevacizumab versus palivizumab plus DMSO, I-BET762, or JQ1 using gene set enrichment analysis (GSEA)[25]. The full details of GSEA across all different comparisons are included in Supplementary Data 5. We also removed all enriched gene sets identified from the comparison between B_DMSO versus P_DMSO to exclude results attributable to bevacizumab treatment alone. Enriched gene sets in P_I-BET or P_JQ1 were identified using an FDR threshold of 0.3 or 0.2, respectively (Fig. 6d). Interestingly, two enriched gene sets which are related to epigenetic regulation of chromosomal structure/activity — signaling events regulated by Class II HDACs identified by Schaefer et al.[26] and genes involved in deposition of new centromere protein (CENP)-A-containing nucleosomes at the centromere described in the peer-reviewed pathway database Reactome — were identified in both P_I-BET and P_JQ1 (Fig. 6d). The enrichment plots of these two gene sets show a similar normalized enrichment score between P_I-BET and P_JQ1 (Fig. 6e), suggesting that genes in these two gene sets were markedly upregulated in palivizumab plus BETi versus bevacizumab plus BETi. In relative terms, they were downregulated in bevacizumab plus BETi.

In summary, the analyses of RNA-Seq data indicate that CDC25B phosphatase activity, Class II HDAC signaling, and deposition of new CENP-A-containing nucleosomes at the centromere may be involved in the altered EC response caused by co-treatment with BETi and bevacizumab.

**Co-treatment with BETi and bevacizumab alters the EC response via CDC25B-regulated chromosomal activity**. Having shown that Class II HDAC signaling and deposition of new CENP-A-containing nucleosomes were upregulated in palivizumab plus BETi, we sought to confirm the involvement of epigenetic regulation in the altered EC response by interrogating these two molecular signaling/processes. Notably, deposition of new CENP-A-containing nucleosomes is mainly controlled by the Mis18 complex (including Mis18α, Mis18β, and M18BP1) and the CENP-A chaperone Holiday junction recognition protein (HJURP) in human cells[27–29]. In addition, we found that cyclin-dependent kinase 1 (CDK1) and CDK2, which are dephosphorylated and activated by CDC25 phosphatases, have been reported to inhibit CENP-A deposition by phosphorylating the Mis18 and/or HJURP complex[30,31]. We, therefore, evaluated whether modulating CDC25 phosphatase activity (as orthogonal testing for CENP-A deposition) or Class II HDAC signaling can change the effects caused by mAb plus JQ1.

To this end, we performed the SMIA for the CDC25 dual specificity phosphatase inhibitor NSC 663284 and the selective Class IIa HDAC inhibitor TMP195. Given that CDC25 phosphatases act as negative regulators for CENP-A deposition, which was upregulated in palivizumab plus BETi and that CDC25B was upregulated in bevacizumab plus BETi, we assessed the effect of CDC25 inhibition on EC survival and/or proliferation with bevacizumab plus JQ1. At three well-separated concentrations tested (1, 3, and 10 μM), the addition of NSC 663284 at a high concentration (10 μM) to co-treatment further reduced the number of viable cells (Fig. 7a), whereas NSC 663284 itself did not alter the EC response to bevacizumab when BETi was not present (Fig. 7b), suggesting that CDC25 activity, as well as CENP-A deposition, were specifically involved in the altered EC response caused by co-treatment with bevacizumab plus BETi. As for Class II HDAC signaling, which was upregulated in palivizumab plus BETi, we assessed the effect of Class IIa HDAC inhibition on EC survival and/or proliferation with palivizumab plus JQ1. Similar to NSC 663284, TMP195 did not change the EC response to bevacizumab when there was no BETi treatment (Fig. 7b). However, none of the three well-separated

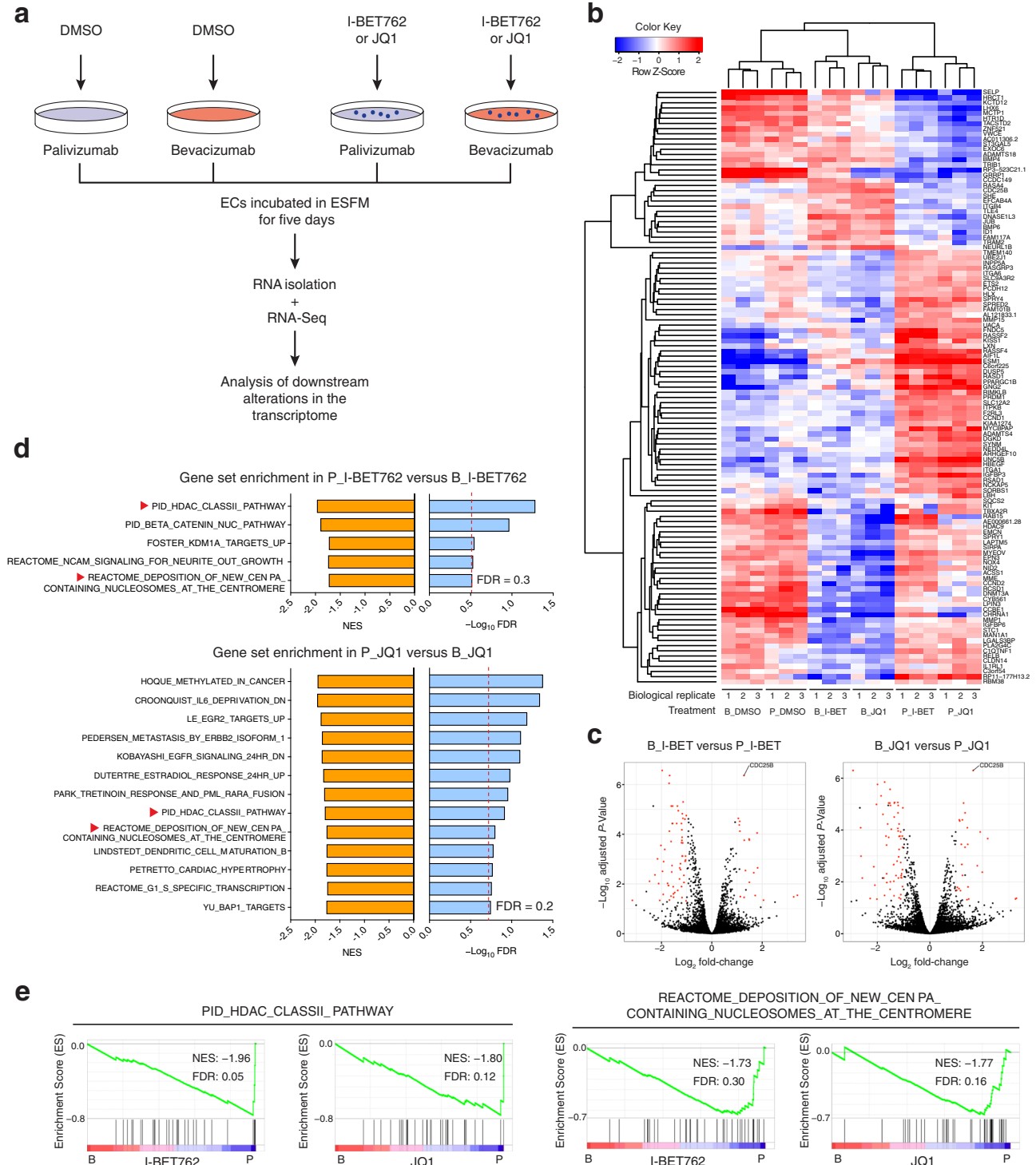

**Fig. 6 Transcriptomic analysis of co-treatment with BETi and bevacizumab in ECs. a** Schematic representation of transcriptomic analysis using RNA-Seq. **b**, **c** Heat map (**b**) and volcano plots (**c**) showing normalized expression dynamics of DEGs identified based on their differential expression (FDR < 0.05 and |log$_2$ fold-change| > 1) in the comparison between bevacizumab-treated versus palivizumab-treated ECs in ESFM plus I-BET762 or JQ1 but not DMSO. DEGs are highlighted in red in volcano plots. **d** Gene sets enriched in ECs treated with palivizumab and I-BET762 (FDR ≤ 0.3) or JQ1 (FDR ≤ 0.2), FDR indicated by red dashed lines. **e** Enrichment plots of the gene set enriched in both conditions of palivizumab plus I-BET762 and palivizumab plus JQ1 (indicated by red arrowheads in **d**). Results represent three independent experiments (n = 3).

concentrations of TMP195 (0.3, 1, and 3 μM) changed the number of viable cells compared with palivizumab plus JQ1 (Fig. 7a).

Collectively, these results suggest the involvement of chromosomal regulation via CDC25B in the altered EC response caused by co-treatment with BETi and bevacizumab.

## Discussion

In this study, we report a pooled 3D CRISPR screen performed in human ECs to identify their response modifiers to bevacizumab, a widely used anti-angiogenic agent that targets the key vascular growth factor VEGFA in human diseases. Among the several

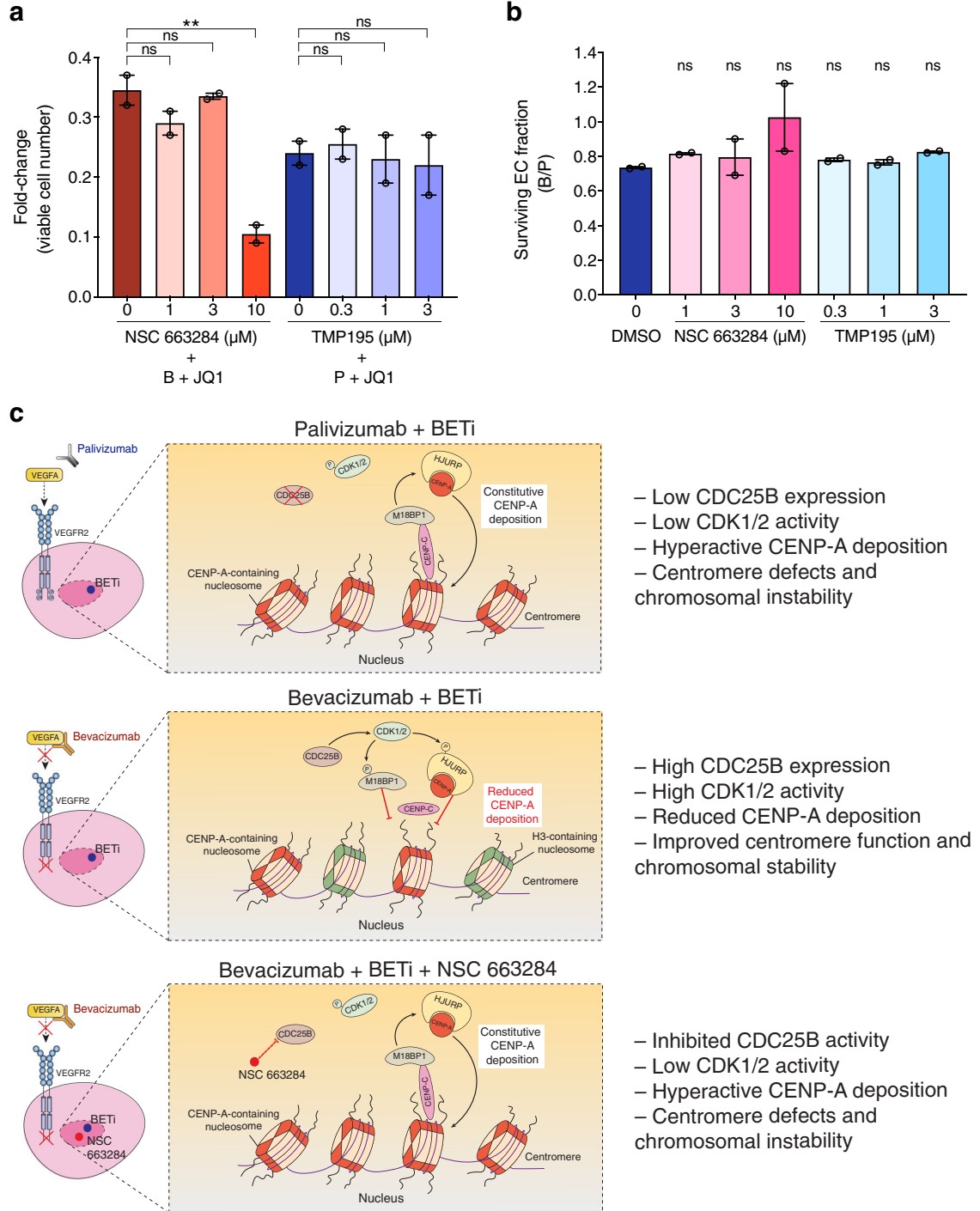

**Fig. 7 CDC25B is associated with the altered EC response caused by co-treatment with BETi and bevacizumab. a** Effects of NSC 663284 or TMP195 on survival and/or proliferation of ECs treated with JQ1 plus bevacizumab or palivizumab in ESFM. The addition of NSC 663284 at 10 μM further reduced viable cell number in ECs treated with JQ1 plus bevacizumab while the addition of TMP195 did not affect those treated with JQ1 plus palivizumab. **b** Relative cell response to bevacizumab in ESFM plus DMSO, NSC 663284, or TMP195. The addition of NSC 663284 or TMP195 did not affect EC response to bevacizumab. Error bars represent ± SEM ($n = 2$ independent experiments). **adjusted $P$ value < 0.01; ns, not significant (versus B + JQ1, P + JQ1 or DMSO, one-way ANOVA with Dunnett's multiple comparison test performed). **c** Schematic model illustrating a potential role for CDC25B in regulating the EC response to co-treatment with BETi plus bevacizumab. Top (palivizumab plus BETi), low CDC25B expression results in phosphorylated cyclin-dependent kinase (CDK) 1/2 and hyperactive centromere protein (CENP)-A deposition, which can cause centromere defects and chromosomal instability. Middle (bevacizumab plus BETi), upregulated CDC25B expression caused by co-treatment dephosphorylates and activates CDK1/2, which reduces CENP-A deposition by phosphorylating the M18BP1 and Holiday junction recognition protein (HJURP) complexes. Eventually, centromere function and chromosomal stability could be improved. Bottom (bevacizumab plus BETi plus NSC 663284), inhibition of CDC25B leads to low CDK1/2 activities in the presence of bevacizumab plus BETi. Reduced CENP-A deposition is therefore restored, which could result in centromere defects and chromosomal instability.

candidates discovered, we demonstrate that functional inhibition of BET bromodomain activity is involved in altering the EC response to bevacizumab through epigenetic regulation of chromosomal activity in a CDC25B-dependent manner. This finding brings together our understanding of the EC response to AAT and epigenetic modulation that modifies chromatin architecture and molecular interactions.

Our application of pooled CRISPR screening to a VEGFA-dependent model in human microvascular blood ECs has allowed the direct and systematic identification of functionally relevant genes and signaling pathways that are responsible for the altered EC response to AAT. This is in contrast to other approaches where the molecular changes are merely correlated with the therapeutic outcomes and may not be directly linked with the cause of the altered response such as resistance. Our model simplifies the heterogeneity of the tumor microenvironment by focusing on one of the major cellular targets of AAT, but incorporates several key factors such as functional VEGFA blockade by bevacizumab and the specialized serum-free medium for optimizing EC survival under VEGFA-dependent culture conditions. In addition, the use of the conditioned medium from a human colon carcinoma cell line LIM1863-*Mph*[32] in the serum-free medium provided a tumor source of the angiogenic secretome and therefore simulated the tumor microenvironment that may have rendered ECs similar to those associated with tumors[33]. Furthermore, the 3D culture system was developed to allow the large-scale in vitro cultivation of blood ECs through dynamic bead-to-bead transfer, which enabled the broad screening of a human kinase-focused library containing genes whose products have already been targeted by many small-molecule inhibitors and biologicals. In particular, this offered an opportunity to evaluate several atypical kinases such as BET proteins which have well-known functions other than phosphorylation.

BET inhibition has been widely studied over the last decade since the discovery of the first BETi JQ1 and I-BET762, especially in the context of preclinical models and clinical trials of different cancer types including hematological malignancies and solid tumors[34–39]. In those studies, inhibition of BET protein functions often results in downregulation of key oncogenes and other genes that are important for regulating cell cycle progression, proliferation, and cell death, thereby causing cancer cell growth arrest and/or apoptosis[34,40,41]. In other cases, BET inhibition has been shown to regulate inflammation, immune response, autophagy, adipogenesis, and cachexia[42–46]. Compared with these studies, research regarding the effect of BET inhibition on endothelial cell biology and angiogenesis is in its infancy. Several recent studies reported an anti-angiogenic effect of BET inhibition using various primary EC culture models and explored its potential in limiting tumor angiogenesis and tumor growth but have yet determined its relationship with anti-angiogenesis[19,20,22,47,48]. In agreement with these findings, our data show that two pan-BETi suppressed key EC activities including survival and/or proliferation, cell cycle progression, and migration while we also disclose a previously unreported mitigating interaction between BET inhibition and VEGFA blockade. Intriguingly, these seemingly conflicting observations may be explained by the recent findings from Gilan et al.[49], which dissected the distinct roles of the two tandem bromodomains (BD1 and BD2) of BET proteins in cancer and immunoinflammation. To illustrate, these authors highlighted that selective inhibition of BD1 primarily altered pre-existing gene expression associated with cell survival and/or proliferation whereas selective BD2 targeting only affected stimuli-induced gene expression[49]. It is, therefore, possible that the suppressed EC activities were due to BD1 inhibition and the mitigating effects on bevacizumab activity were caused by BD2 inhibition when BD1 and BD2 were both inhibited by JQ1 or I-BET762 in our system.

This could lead to an interesting hypothesis to be tested in the future that selective inhibition of BD2 but not BD1 mediates EC resistance to bevacizumab. Moreover, the DEG analysis and GSEA of RNA-Seq data show concordant results generated by co-treatment with bevacizumab and JQ1 or I-BET762. Together with the validation using the CDC25 inhibitor, our results reveal a potential role for CDC25B phosphatase in the altered EC response to bevacizumab and BETi through regulating the chromosomal activity of CENP-A deposition (Fig. 7c). Indeed, constitutive CENP-A deposition throughout the cell cycle has been associated with severe defects of chromosome segregation and multipolar spindles during mitosis and may adversely affect centromere function and genomic integrity in human cells[50]. This is in agreement with our observations that upregulated *CDC25B* expression and reduced CENP-A deposition were associated with the improved EC survival and/or proliferation (the resistant phenotype) under co-treatment with BETi and bevacizumab.

Although tumors comprise multiple cell types, the molecular interactions within a non-mutated cellular component such as ECs, which are exposed to the dynamic inputs and outputs in the tumor microenvironment may provide informative readouts for predicting and/or monitoring the whole tumor response[8]. Clinically, tumor ECs or those purified from the circulation of patients might serve as a convenient diagnostic tool for assessing the therapeutic response, which would become more feasible with the emerging application of single-cell transcriptomic analysis[11]. Furthermore, although we used an immortalized microvascular blood EC line that is highly biologically relevant and maintains key features of primary blood ECs, further validation and investigation in primary human samples are warranted to gain clinical impact. With the newly developed clinical databases revealing EC-specific data from single-cell sequencing, the EC response modifiers, DEGs and the relevant pathways identified in this study could offer useful information about strategies for overcoming resistance and developing potential biomarkers for AAT, BETi, and/or combination therapy in a range of disease contexts.

Overall, our findings provide an entry point for understanding EC response to AAT and can eventually enhance the effectiveness of this therapeutic approach. Angiogenesis-dependent diseases such as cancer and eye conditions could be reprogramed to respond differently to ATT by modulating the epigenetic activity within ECs and directing subsequent vessel remodeling. Further understanding of such mechanisms will provide additional benefits for improving the use of AAT and/or enhancing the efficacy of other treatment options, for example, immunotherapy where a relationship between vessel normalization and immunostimulatory reprograming has already been established[51].

## Methods

**Proteins and chemicals**. Bioactive low-endotoxin recombinant human vitronectin (rhVTN) was produced as described[52]. The pan-BETi JQ1 and I-BET762 (Selleckchem or MedChemExpress), the small-molecule TLK1 inhibitor thioridazine HCl (THD; Sigma), the small-molecule CDC25 dual specificity phosphatase inhibitor NSC 663284 (MedChemExpress), and the small-molecule selective Class IIa HDAC inhibitor TMP195 (MedChemExpress) was reconstituted and stored according to the manufacturer's instructions. The small-molecule thousand-and-one amino acids protein kinase (TAOK) inhibitor, *N*-[2-oxo-2-(1,2,3,4-tetra-hydronaphthalen-1-ylamino)ethyl]biphenyl-4-carboxamide (referred to as compound 43)[53], was synthesized by SYNthesis med chem (Parkville, VIC, Australia), dissolved in DMSO, aliquoted and stored at −20°C under N[2]. Bevacizumab and palivizumab were from Roche and MedImmune, respectively.

**Cell lines and monolayer culture**. The immortalized cell line (XSEB113C1, Lonza) was generated from primary human female dermal microvascular blood ECs by transduction with a γ-retrovirus expressing the human telomerase reverse transcriptase catalytic subunit by the manufacturer. This XSEB113C1 cell line has been validated by the manufacturer for common blood EC markers and characteristics of primary cells (e.g., these cells are responsive to VEGFA, can form tubes, and have a finite lifespan in vitro). ECs were maintained in EGM-2MV medium

prepared from endothelial cell basal medium-2 (EBM-2; Lonza) supplemented with EGM-2MV BulletKit (Lonza). If not mentioned otherwise, cell culture vessels for monolayer culture of ECs were briefly incubated with 5 µg/mL human fibronectin (BD Biosciences) then air-dried at room temperature. ECs were incubated in 5% $CO_2$, 5% $O_2$, and 90% $N_2$ at 37°C. The human embryonic kidney 293 T (HEK293T; Open Biosystems) cell line was maintained in D10 medium prepared from 1× Dulbecco's Modified Eagle Medium (DMEM; Life Technologies) supplemented with 10% v/v FBS (SAFC Biosciences), 1 mM sodium pyruvate (Life Technologies), 2 mM GlutaMax-I (Life Technologies), penicillin–streptomycin (100 U/mL, 100 µg/mL, respectively; Life Technologies) in 10% $CO_2$ at 37°C. For detachment, ECs were treated with Accutase (Sigma) at 37°C for 8 min and HEK293T cells were treated with trypsin-EDTA. For cell counting with high precision, cells were pelleted at $250 \times g$, room temperature for 5 min, and the supernatant was aspirated. The cell pellet was resuspended in diluent [20% v/v flow buffer (1× PBS, 20 mM HEPES (Life Technologies), 0.5% w/v bovine serum albumin (BSA; Sigma), 0.5 mM EDTA, pH 7.4 at 23°C), 80% v/v AccuMax (Sigma), EDTA at a final concentration of 2.5 mM]. The cell suspension was mixed in 96-well plates with the pro-fluorescent stains calcein violet-acetoxymethyl ester (final concentration, 160 nM; Life Technologies) and SYTOX Red (final concentration, 5 nM; Life Technologies) in each well. The plates were incubated at room temperature in the dark for 15 min then analyzed by volumetric flow cytometry (FACSVerse flow cytometer, BD Biosciences). The total number of events acquired per sample was 10,000. Data acquisition was performed using FACSuite software (BD Biosciences) and analysis was performed using FlowLogic (version 600.0 A; Inivai Technologies) or FlowJo software (version 10.0.8r1; FlowJo). Otherwise, cells were counted using an imaging-based benchtop assay platform (Countess II automated cell counter, Life Technologies). All human cell lines were verified free of mycoplasma infection.

**Microcarrier-based cell culture**. Plastic microcarriers (SoloHill Engineering, Pall Corporation, #P-221-050; referred to hereafter as SoloHill Plastic microcarriers or SPM) were coated with 0.1 mL/cm² of 5 µg/mL rhVTN in PBS at 4°C overnight with continuous gentle mixing. The next day, the coated SPM were washed with a warm seeding medium containing EBM-2 supplemented with 0.2% FBS (Lonza), 75 µM L-ascorbate ((+)-sodium L-ascorbate; Sigma)/500 µM 2-phospho-L-ascorbic acid (2-phospho-L-ascorbic acid trisodium salt; Sigma) pH 7.4, 10 µg/mL gentamicin (Life Technologies), 1 µg/mL hydrocortisone (Sigma) and 20 mM HEPES (Life Technologies). The washed SPM were resuspended in seeding medium at 40 mg/mL (equivalent surface area, 14.4 cm²/mL) and aliquots were added to 125 mL disposable spinner flasks (referred to as 125 mL SFs; Corning, #3152). On day 0, ECs were used to inoculate microcarriers at 7000 cm⁻² in seeding medium while stirring at 67 rpm. After 1 h of incubation at 37°C in 5% $CO_2$, 5% $O_2$, and 90% $N_2$ with stirring, a master mix of serum and growth factors (prepared from an EGM-2MV BulletKit) was added to the culture. The final FBS and GFs concentrations were 5% v/v and 1×, respectively. The culture was then incubated at 37°C in 5% $CO_2$, 5% $O_2$, and 90% $N_2$ with stirring for 17 h. The next day (day 1, 18 h after inoculation), FBS was added to the culture in EGM-2MV to an extra 10% to achieve a final FBS concentration of 15% (referred to as microcarrier growth medium). Samples were taken for counting nuclei and for fixation, staining, and imaging. Meanwhile, the stirring was changed from constant to intermittent: a 3 min on/30 min off (i.e., stirring for 3 min then stopping for 30 min) cycle for 6 h. After 6 h, intermittent stirring was changed to an 11 h on/1 h off cycle. Intermittent stirring (11 h on/1 h off) was used throughout the culture period after day 1 unless mentioned otherwise. Samples were taken on day 1 and every 48 h thereafter to monitor cell number and distribution. Fifty percent of microcarrier growth medium was changed on day 2 and every subsequent 48 h. Nuclei were released, counted, and analyzed according to He et al.[54] To illustrate, after microcarriers in a sample sedimented, the supernatant was removed and the microcarriers were washed with PBS twice. Nuclei were released using cell lysis solution (0.1 M citric acid (Sigma), 1% v/v IGEPAL CA-630 (Sigma)) with vortexing and the microcarriers were then removed by applying the microcarrier slurry to a cell sieve (pore size, 70 µm) followed by a spin at $150 \times g$, room temperature for 1 min. The nuclei suspension was collected and the volume was measured. If not analyzed immediately, the nuclei suspension was stored at 4°C for up to 2 weeks. Nuclei counting was performed by volumetric flow cytometry (FACSVerse flow cytometer). Nuclei suspension was mixed with the pro-fluorescent nucleic acid stain SYTOX Red (final concentration, 5 nM). The total number of events acquired per sample was 20,000. Data acquisition was performed using FACSuite software and analysis was performed using FlowLogic or FlowJo software. For fixation, staining, and imaging of nuclei on microcarriers, cells were fixed in 2% paraformaldehyde and stained with 1 µg/mL Hoechst 33342 (Life Technologies) at 37°C for 20 min. After one wash with PBS, the stained nuclei were imaged using epifluorescence.

**siRNA transfection**. Transfection of ECs for delivering siRNA was performed using a cationic lipid-mediated transfection reagent (Lipofectamine RNAiMAX transfection reagent, Life Technologies) according to the manufacturer's instructions. siRNAs targeting specific genes were purchased as siGENOME siRNA SMARTpools from Dharmacon (each SMARTpool is an equimolar mix of four siRNAs targeting the same gene provided as a single reagent). The ON-TARGETplus Non-targeting Control Pool (referred to as OTP-NT; each pool contained an equimolar mix of four non-targeting siRNAs; Dharmacon) was used as a negative control. All siRNA pools were used at 20 nM unless specified otherwise.

**Lentivector preparation and transduction**. To prepare lentivectors (LV), HEK293T cells were used to seed T225 flasks (Corning) at $1 \times 10^5$ cm⁻² in LV collection medium containing Advanced DMEM (Life Technologies) supplemented with 2% FBS (SAFC), 0.01 mM L-α-lecithin (Sigma), 1× CD lipid concentrate (Life Technologies), 0.01 mM cholesterol (Sigma) and 2 mM GlutaMax-I (Life Technologies). The next day (day 1, 24 h after seeding), cells were co-transfected with the LV transfer plasmid(s), the LV packaging plasmid pCD/NL-BH*ΔΔΔ (Addgene, #17531) and the envelope-encoding plasmid pLTR-G (Addgene, #17532) at a mass ratio of 2:1:1, respectively, using polyethylenimine (PEI)-mediated transfection[55]. Sixteen hours after transfection (day 2), all supernatant in the T225 flasks was removed, disinfected, and discarded then replaced with a reduced volume of warm LV collection medium. This was the first medium change. Twenty-four and 48 h after the first medium change (days 3 and 4, respectively), all medium supernatant was collected and filtered. On day 4, the pooled day 3 and day 4 filtrates containing the LV were gently and thoroughly mixed with 0.25× volume of 50% sterile poly(ethylene glycol) with an average molecular weight of 8000 (PEG8000; Sigma) in PBS. The LV–PEG8000 mixture was incubated at 4°C overnight. The next day (day 5), the LV–PEG8000 mixture was centrifuged and the LV pellets were gently resuspended in cold sterile PBS plus 1% BSA (Probumin, Merck Millipore). The LV suspension was then gently and thoroughly mixed by pipetting, aliquoted, frozen on crushed dry ice, and stored at −80°C. ECs were transduced with LV in culture medium plus 0.075% Pluronic F-127 for 16 h and selected with puromycin (Life Technologies) 48 h after the beginning of transduction for three days to kill non-transduced cells. Stirring during the period from the beginning of transduction to the completion of puromycin selection was changed from intermittent to constant stirring, and changed back to intermitted stirring (11 h on/1 h off) afterward.

**Kinome-wide CRISPR screen**. ECs were incubated for expansion and monitored in a total number of two 125 mL SFs under identical conditions. Pooled transduction of cells with the single-vector Cas9/sgRNA library targeting the human kinome[56] (referred to as hCKL for human CRISPR kinome library; Addgene, #75314) was performed when the cell density reached $1.2–1.5 \times 10^4$ cm⁻² at a multiplicity of infection (MOI) of 1–1.5 (with an average coverage of at least 2000 cells per sgRNA; day −5). Twenty-four hours after the beginning of transduction (day −4), the culture medium was removed as much as possible and replaced with the same volume of warm microcarrier growth medium. Forty-eight hours after the beginning of transduction (day −3), 0.5 µg/mL puromycin was added to the culture to kill non-transduced cells for three days. During this period, a stepwise reduction of serum concentration was also applied as follows. Twenty-four hours after the addition of puromycin (day −2), the culture medium (15% FBS) was removed as much as possible and replaced with the same volume of warm EGM-2MV (5% FBS) plus 0.5 µg/mL puromycin. Twenty-four hours after the last medium change (day −1), the culture medium was removed as much as possible and replaced with the same volume of warm BCM-2+ (1% FBS; Supplementary Table 1) plus 0.5 µg/mL puromycin. Twenty-four hours after the last medium change (day 0), the cultures from two SFs were combined to be washed gently with MCA wash solution (Supplementary Table 1) twice then transferred 1:1 back into the two original SFs. A final wash was performed using an ESFM for microcarrier-based culture plus the anti-VEGFA mAb bevacizumab or the isotype-matched control mAb palivizumab (ESFM-3D + mAb; Supplementary Table 2). Subsequently, the volume of the culture in each SF was adjusted by adding ESFM-3D + mAb for a final microcarrier concentration of 40 mg/mL. The cultures were then maintained in ESFM-3D + mAb for 21 days (with 50% of the medium changed every 48 h) and were monitored by nuclei counting every 72 h. On days 0, 12, and 21, nuclei were released from a sample of the microcarrier-based cultures and stored at 4°C before gDNA isolation. The number of nuclei required per sample was calculated to maintain coverage of 2000 cells per sgRNA at each time point (with the exception of the day 0 sample, 1000 cells per sgRNA) to enable detection of depleted sgRNAs during screening selection. Genomic DNA from ECs on microcarriers was isolated using anion-exchange chromatography (Blood and Cell Culture DNA Mini Kit, QIAGEN) according to the manufacturer's instructions with subtle modifications as follows. In the samples taken from the kinome-wide CRISPR screen, nuclei were pelleted at $1500 \times g$, 4°C for 15 min. After removal of the supernatant, nuclei pellets were resuspended in Buffer G2 from the kit and all following steps were performed according to the manufacturer's instructions. The sgRNA-encoding cassettes were amplified by PCR using isolated gDNA or purified pDNA as templates. The sequences of the forward (i.e., P5 XPR/LKP1 primer mix, an equimolar mixture of eight single P5 primers) and reverse (i.e., P7 XPR023 with a unique P7 barcode sequence for each primer) primers are listed (Supplementary Table 5). For PCR amplification, each 50 µL reaction contained up to 2.5 µg gDNA or 1 ng pDNA, 1 µM P5 primer mix, 1 µM P7 primer, and 1× NEBNext Ultra II Q5 Master Mix (NEB). The following thermocycling parameters were used: 98°C for 30 s, 16 cycles of (98°C for 10 s, 65°C for 75 s), 65°C for 5 min, then held at 4°C. The PCR products were purified using silica-membrane technology (QIAquick PCR Purification Kit, QIAGEN). The purified PCR products were analyzed by electrophoresis (High Sensitivity D1000 ScreenTape assays, Agilent Technologies) with an

analyzer (Agilent 4200 TapeStation instrument, Agilent Technologies) for fragment and concentration measurement. DNA size selection for the amplified sgRNA-encoding cassettes was performed using electrophoresis in agarose gels (Pinpin Prep, Sage Science) if necessary. Subsequently, the samples were purified using magnetic beads (Agencourt AMPure XP, Beckman Coulter). The purified samples were sequenced on a NextSeq instrument (Illumina) with a single-end 75 bp run using the high-output mode.

**CRISPR screen analysis**. Raw reads generated from the NextSeq platform were quality checked using FastQC[57] and de-convoluted for assignment to different conditions of the screen according to the barcode included in the P7 primers. The aligned reads were trimmed to remove the 5′ adapter sequence (by searching for the common 5′-CACCG-3′ sequence) using Cutadapt (version 1.13)[58]. After trimming, the next 20 nt sequences represented the sgRNA insert. Median normalization and alignment of the reads to the hCKL were performed using the statistical package Model-based Analysis of Genome-wide CRISPR/Cas9 Knockout (MAGeCK; version 0.5.7)[24]. In brief, enrichment and depletion of sgRNAs were analyzed by comparing the normalized reads of each sgRNA in the bevacizumab-treated versus palivizumab-treated samples at each respective time point. Identification of candidate genes was performed using the sgRNA ranking results with a modified α-RRA algorithm and multiple comparisons.

**EC-MCA**. ECs were transduced at an MOI of 0.3–0.4 with LV encoding tGFP or DsRed-Max (referred to as XSEB-tGFP or XSEB-DsRed-Max cells, respectively). XSEB-tGFP cells were transfected with gene-specific siGENOME SP siRNAs to generate experimental cells and XSEB-DsRed-Max cells were transfected with a non-targeting siRNA pool to generate negative control cells. Twenty-four hours after the beginning of transfection, green experimental and red negative control cells were harvested, counted, and mixed at a ratio of 1:1. The mixed cell population was used to seed rhVTN-coated six-well plates in BCM-2+ at 12,000–14,000 total cells cm$^{-2}$. Approximately 20 h later (day 0), BCM-2+ was aspirated and the wells were gently washed with warm MCA wash solution. Subsequently, cells were incubated with ESFM-2D + mAb (Supplementary Table 3). After nine days of incubation in 5% O$_2$, 5% CO$_2$, and 90% N$_2$ at 37°C with medium changed every 48 h, cells were harvested and analyzed by flow cytometry. A final EC-MCA ratio ($M$) was calculated based on the green-to-red viable cell ratios using the formula:

$$M = \frac{(green/red)_B}{(green/red)_P} \tag{1}$$

where $B$ or $P$ represents the inclusion of bevacizumab or palivizumab, respectively.

**SMIA**. The SMIA is conceptually similar to that of the EC-MCA except that wild-type cells treated with small molecules or vehicles were incubated separately and the cell number in each well was determined by counting released nuclei on day 9. Specifically, ECs were used to seed rhVTN-coated 6- or 12-well plates at 4000 cm$^{-2}$ in BCM-2+. ESFM-2D + mAb plus a small-molecule inhibitor or vehicle was added to the wells after washing with MCA wash solution on day 0 and then changed every subsequent 48 h. After nine days of incubation, nuclei were released and counted. The nuclei number from each well was used to calculate a final SMIA ratio ($S$) using the formula:

$$S = \frac{(drug/vehicle)_B}{(drug/vehicle)_P} = \frac{(B/P)_{drug}}{(B/P)_{vehicle}} \tag{2}$$

where $B$ or $P$ represents the inclusion of bevacizumab or palivizumab, respectively.

**Protein extraction, antibodies, and immuno blotting**. Whole-cell protein was extracted in 1× lithium dodecyl sulfate sample buffer (Life Technologies), prepared with 25 mM Tris (2-carboxyethyl) phosphine hydrochloride (TCEP-HCl; Thermo Fisher Scientific), separated by sodium dodecyl sulfate-polyacrylamide gel electrophoresis, and probed with primary antibodies (mouse monoclonal anti-α-tubulin, Santa Cruz, #sc-8035, 1:200; rabbit monoclonal anti-GAPDH, Cell Signaling Technology, #2118, 1:2000; mouse monoclonal anti-FLAG M2, Sigma, #F3165, 1:5000).

**Assay evaluating cell survival and/or proliferation**. For evaluating the effect of siRNA in a complete growth medium, ECs were transfected 24 h before used to seed six-well plates at 4000 cm$^{-2}$ in EGM-2MV on day 1. On days 2, 3, 5, 7, 9, and 11 after the beginning of transfection, cells in the wells were imaged using phase-contrast microscopy then nuclei were released using cell lysis solution and counted using volumetric flow cytometry (FACSVerse flow cytometer). For evaluating the effect of BETi in a complete growth medium, ECs were used to seed six-well plates at 4000 cm$^{-2}$ in EGM-2MV. EGM-2MV containing JQ1 at 300 nM, I-BET762 at 1000 nM, or vehicle (0.1% v/v DMSO) was added ~20 h after seeding (day 0) and changed every subsequent 48 h. On days 0, 1, 3, 5, 7, and 9 of BETi treatment, cells in the wells were imaged then nuclei were released and counted. For evaluating the interaction between BETi and bevacizumab, ECs were used to seed rhVTN-coated six-well plates at 4000 cm$^{-2}$ in BCM-2+. Approximately 20 h later (day 0), BCM-2+ was aspirated and the wells were gently washed with warm MCA wash solution.

EGM-2MV/ESFM-2D/ESRM (Supplementary Table 4) + mAb plus DMSO or BETi (I-BET762, 1000 nM; JQ1, 300 nM) was added to the wells after washing with MCA wash solution on day 0 and changed every subsequent 48 h. On days 0, 1, 3, 5, 6, 7, and 9 of BETi treatment, cells in the wells were imaged and nuclei were released and counted. For quantification, the nuclei number per well per condition at a specific time point was normalized to the nuclei number per well on day 0.

**Nuclear DNA content analysis**. ECs were used to seed 10 cm dishes at 4000 cm$^{-2}$ in EGM-2MV. EGM-2MV containing JQ1 at 300 or 1000 nM, I-BET762 at 1000 nM or 0.1% v/v DMSO was added ~20 h after seeding (day 0). Nuclei were released after three days of incubation and were used for the analysis of nuclear DNA content according to He et al.[54]. To illustrate, nuclei were pelleted at 1500 × $g$, 4°C for 10 min, resuspended in PBS and treated with 0.1 µg/mL DNase-free RNase A (Life Technologies) for 30 min at room temperature before mixing with propidium iodide (final concentration, 5 µM; Life Technologies). Samples were acquired by flow cytometry (FACSVerse flow cytometer). The total number of single nuclei acquired per sample was 10,000. Data acquisition was performed using FACSuite software and analysis was performed using FlowLogic or FlowJo software.

**Scratch wound migration assay**. ECs were used to seed 96-well flat clear-bottom black-walled polystyrene microplates (Corning) at 15,000 per well in EGM-2MV. EGM-2MV containing DMSO or BETi (I-BET762, 1000 nM; JQ1, 300 nM) was added ~20 h after seeding. Cells were incubated for another 48 h. Subsequent staining with 5-chloromethylfluorescein diacetate (CellTracker Green CMFDA dye, final concentration, 5 µM; Life Technologies) for 20 min at 37°C, scratching, washing using EGM-2MV (fresh EGM-2MV plus DMSO, I-BET762 or JQ1 was added after washing), and imaging was performed at 0 h (T0). Precise scratching (0.38 × 3.8 mm) was performed using a wounding replicator with FP pins (V&P Scientific) driven by a workstation robotic liquid handler (Sciclone ALH 3000 workstation, Caliper Life Sciences). Fluorescent images of cells were captured using a high-throughput imaging system (Cellomics ArrayScan VTI; Thermo Fisher Scientific) and data acquisition was performed using high-content acquisition software (Cellomics Scan Software, version 7.6.2.1; Thermo Fisher Scientific). Fixation using 4% paraformaldehyde for 20 min at room temperature, actin staining with phalloidin CF488A (final concentration, 2 U/mL; Biotium), nuclear staining with Hoechst 33342 (final concentration, 2 µg/mL) for 1 h at room temperature, washing with PBS and imaging were performed 24 h after scratching (referred to as T24). Images were processed using CellProfiler (version 2.2; Broad Institute) and MetaMorph (version 7.10.3; Molecular Devices) and cell number under different conditions was quantified using ImageJ (version 2.0.0). Relative EC migration was calculated using the formula:

$$\text{Relative EC migration (\%)} = \left(\frac{A0 - A24}{A0}\right) \times 100 \tag{3}$$

where $A0$ is the scratch area at T0 and $A24$ is the scratch area at T24.

**RNA-Seq and analysis**. Total RNA was isolated using silica-membrane technology (RNeasy Plus Mini Kit, QIAGEN) and treated with RNase-free DNase (TURBO DNA-free Kit, Life Technologies) according to the manufacturer's instructions. For assessing baseline gene expression, ECs were incubated in EGM-2MV for five days before RNA isolation. For assessing the effects of co-treatment with BETi and bevacizumab, ECs were incubated in ESFM-2D + mAb plus DMSO or BETi (I-BET762, 1000 nM; JQ1, 300 nM) for five days before RNA isolation. RNA-Seq library was prepared using total RNA, reverse transcriptase, and DNA polymerase (QuantSeq 3′ mRNA-Seq Library Prep Kit FWD for Illumina, Lexogen, Vienna, Austria, #015.96). The samples were sequenced on a NextSeq instrument with a single-end 75 bp run using the high-output mode. Alignment of the reads to a reference genome (human HG19) was performed using HISAT2 (version 2.1.0)[59]. Transcripts and frequencies were determined from the aligned reads using featureCounts[60] or HTSeq (version 0.10.0)[61]. Differential gene expression analysis was performed using DESeq2 (version 3.5)[62] or the edgeR/voom–limma workflow[63–65]. GSEA was performed using the Molecular Signatures Database (MSigDB; with the C2 collection)[25]. Mean-centering and unsupervised hierarchical clustering on both samples and genes were applied. The distance between the clusters was calculated according to Euclidean distance (on sample-wise comparisons) or correlation (on gene-wise comparisons).

**Statistics and reproducibility**. All statistical analyses (except the analyses of deep sequencing and RNA-Seq) were performed using Prism 9 software (GraphPad Software Inc.). Only the results obtained from independent experiments ($n \geq 2$) were included in statistical analysis. All error bars represent ± standard error of the mean (SEM). For analysis of EC-MCA and SMIA ratios (i.e., $M$ and $S$, respectively), data were first log$_2$-transformed because the assumption was made that log$_2 M$ and log$_2 S$ were normally distributed. Log$_2$ transformation was also applied to ratios used in the analysis of the surviving EC fraction and fold-change in viable cell numbers. For RNA-Seq analysis of ECs treated with DMSO or BETi (I-BET762 or JQ1) in ESFM plus bevacizumab or palivizumab, an initial list of DEGs between bevacizumab plus I-BET762 or JQ1 versus palivizumab plus I-BET762 or JQ1 was generated from the RNA-Seq analysis of three independent experiments using the

criteria: FDR < 0.05 and |log$_2$ fold-change| > 1. To ensure DEGs were specific to the comparison between bevacizumab plus BETi versus palivizumab plus BETi, those that appeared in the comparison between bevacizumab plus DMSO versus palivizumab plus DMSO were further excluded to obtain the final list of DEGs. Statistical significance was determined by a two-tailed unpaired Student $t$ test for two-group comparison or one-way analysis of variance (ANOVA) for multiple comparisons (Dunnett's multiple comparison test performed with adjusted $P$ values reported). For all statistical analyses, a $P$ value or adjusted $P$ value < 0.05 was considered sufficient to reject the null hypothesis.

**Reporting summary**. Further information on research design is available in the Nature Research Reporting Summary linked to this article.

## Data availability

The RNA-Seq data set has been deposited in Gene Expression Omnibus (GEO) with the accession number GSE176149. Relevant data are available in the article and Supplementary Information. Source data for the RNA-Seq analysis (Fig. 3a; Fig. 6b–e and Supplementary Fig. 2b) are available in Supplementary Data 1, 4, and 5. Source data for the CRISPR screen analysis (Fig. 2d, e) are available in Supplementary Data 2 and 3. Source data for the other figures (Fig. 1c; Fig. 2b, c; Fig. 3d, f; Fig. 4a–c, e, f; Fig. 5b–g; Fig. 7a, b; Supplementary Fig. 3d, e and Supplementary Fig. 5b, c) are available in Supplementary Data 6. Additional relevant data are available from the corresponding author upon reasonable request.

## Code availability

No custom code, algorithm, or software has been used during this study. Please see Methods and Supplementary Methods for descriptions in detail.

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

## Acknowledgements

We thank the following Peter MacCallum Cancer Centre core facilities: Ralph Rossi and staff at Flow Cytometry, Gisela Mir Arnau and staff at Molecular Genomics, and Jason Li and staff at Bioinformatics, for technical support and assistance on data analysis. This study was supported by Program Grants from the National Health and Medical Research Council of Australia (NHMRC) and by funds from the Operational Infrastructure Support Program provided by the Victorian Government, Australia. S.A.S. and M.G.A. are supported by a Senior Research Fellowship from the NHMRC. M.Y.H. was supported by a University of Melbourne Postgraduate Scholarship from the University of Melbourne, Australia; Z.L.G. by an Australian Government Research Training Program Scholarship and L.C. by the L.E.W. Carty Charitable Fund.

## Author contributions

M.Y.H., M.M.H., M.G.A., and S.A.S. conceived the study. M.Y.H., M.M.H., and S.A.S. designed and planned experiments. M.Y.H. performed experiments. M.Y.H., M.M.H., and S.A.S. analyzed and interpreted data with contributions from R.L., J.P.R., Z.L.G., L.C., O.G., Y.-C.C., M.A.D., and M.G.A. R.L. assisted in performing and analyzing results from scratch wound migration assay. N.T. performed computational analysis of RNA-Seq data and data visualization. M.Y.H. and S.A.S. wrote the manuscript. All authors contributed to the editing of the manuscript and approved the final version.

## Competing interests

S.A.S. and M.G.A. are shareholders of Opthea Ltd., a company involved in developing therapeutics and diagnostics for vascular targets. The remaining authors declare no competing interests.
