## [Peer Review File · Communications Biology]

Reviewers' Comments:

Reviewer #1:

Remarks to the Author:

The manuscript by He et al. describes a highly novel study to investigate and identify mechanisms by which endothelial cells overcome growth inhibition mediated by the anti-angiogenic agent bevacizumab. This study uses a range of cutting edge technologies and approaches which are applicable to wide range of biological questions beyond this study of resistance to vascular endothelial growth factor (VEGF)-targeted anti-cancer therapy.

The authors have developed a system in which they are able to culture endothelial cells at high density allowing a kinome-wide CRISPR-Cas9 screen to be undertaken. Cells that preferentially grew in the presence of anti-VEGF antibodies were analysed to determine which kinases had been targeted, and candidate kinases were validated using siRNA mediated knockdown experiments. Some of these kinases included members of the bromodomain and Extra-Terminal motif (BET) protein family involved in chromatin regulation. The use pharmacological inhibitors to these proteins enabled confirmation of the role of these kinases in mediating resistance to anti-VEGF treatment. The paper explored how inhibiting these proteins affected endothelial proliferation, motility and cell cycle progression and sought to understand using next generation sequencing how inhibiting these BET proteins enabled endothelial cells to overcome VEGF blockade. This resulted in the identification of genes involved in modulating chromatin activity and these findings were again interrogated using small molecular inhibitors targeting class II histone deacetylases and deposition of new centromere proteins. The greatest strength of this paper is its experimental approach to identify kinases involved in a clinically relevant biological question, and then to further use next generation sequencing to identify how they might be acting. Its weakness is that the findings have not been explored in any detail and this is acknowledged in the manuscript. This would involve considerably more work that is beyond the scope of this study. This study contributes to our understanding of our understanding of how endothelial cells may adapt to VEGF targeted therapy.

Overall this is an excellent piece of work and the wider applicability of the approaches to many areas of biology make this highly suitable for publication in Communications Biology.

Some more specific recommendations/ comments

In the results commentary the cells used are described as endothelial cells, some reference is made to immortalisation in the legend of figure 1 and indeed in the Materials and Methods section it notes that the cells used are an immortalised cell line derived originally from dermal microvascular cells. Typically, studies with endothelial cells use primary cells and so it would be good to be clearer on this point particularly in the main results commentary. While it understandable that an immortalised cell line would be necessary for the CRISPR-Cas9 screen, the other studies, such as the siRNA transfection confirmatory experiments, should ideally be done with primary endothelial cells (to ensure that the findings from the first screen were not related to endothelial cell transformation). Clarification of the cells used throughout would be useful and if the immortalised cell line was used throughout, this should be acknowledged and the associated limitations discussed.

This is a complex study and the authors have included a number of useful schematics which aid the reader in understanding the various experimental protocols. The reader would be additionally helped if the figure legends were more informative and summarised the findings demonstrated in the figure, and if all subsections of the results had a summary at the end clearly explaining the findings of that section (this is there for many but not all sections). A flow chart of the entire study would also be useful.

As described in the methods sections the statistical approaches are sound, however they are not always applied in the way outlined in this section. For example, in figure 3d which contains multiple

groups, t-tests rather than ANOVA were performed. In this particular study this is an additional problem in that the ratios generated with the different knockdowns are compared to 1, however this does not account for any variation that one would expect to see in a control sample where both sets of green and red fluorescent protein expressing cells are transfected with control siRNA duplex. Such a control for this experiment is not evident from the data, and it is important both for the statistical analyses and for ruling out the theoretical possibility that expression of different fluorescent proteins differentially affects the cell proliferation.

Reviewer #2:

Remarks to the Author:

In this manuscript, the authors describe an experimental analysis aimed at uncovering the mechanisms of resistance to anti-angiogenesis therapy. More specifically, they have assessed the pathways that underly resistance to the anti-VEGFA antibody bevacizumab and have done so using a CRISPR-Cas9 loss-of-function screen of the kinome and in vitro, 3D microcarrier-based culture of human microvascular endothelial cells. The major claim is that epigenetic reader proteins of the BET family can in part explain how MVECs respond to bevacizumab treatment.

Comments:

1. This analysis is one of very few to make progress in defining the mechanisms that underly resistance to anti-angiogenic treatment. This is a thorny issue because it is complex and because there are real-world implications for cancer treatment. This subject matter is thus of considerable importance and appropriate for a high-level journal.
2. This is an impressive study because the analysis is innovative, meticulous and by-and-large, thorough. Though this is a study based on culture analysis, it is well thought out and establishes a launching point for future in vivo experimentation that can assess the role of BET protein activities in resistance to anti-angiogenic therapy.
3. In Figure 3f, a no-drug control is missing. Where you have a nice dose response (as with compound 43) it may be a little academic, but with one of the compounds (JQ1) used throughout the rest of the analysis, there is no dose response, and so a no-drug control is quite important. This should be included for all f panels of this figure.
4. Please define the culture conditions for the experiments described in Figure 4. Obviously, the reader is anticipating an analysis of VEGFA-dependent culture conditions, but it isn't clear whether this is the case for the data in this figure.
5. The analysis described in Figure 5 appears to be valuable but is complex and it isn't always clear exactly what was done. My understanding is that all the data in panels b-e of Figure 5 were generated using the cell death assay described in Fig. 3e. If this is correct, please make sure this is explicitly stated. If this isn't correct, then I have nicely illustrated the problem for you.
6. Is there an assay for the BET proteins that more directly assesses their activity? As it stands, we don't have confirmation of this in these assays and we have to make the assumption that the inhibitors perform as advertised. Can you use a population of MVECs that are CRISPR targeted for BET gene loss-of-function as an adjunct to the pharmacological assays? The inhibitor assays are valuable in their own right, but if they were bolstered by genetic loss of functions (assays with similar responses) this would add a level of certainty to the interpretation.
7. In this sentence, "Noticeably, deposition of new CENP-A-containing nucleosomes...., The first word should be "Notably, .."

8. Finally, the way this manuscript is written tends to make it somewhat opaque. There are many abbreviations and quite a bit of convoluted sentence structure. If the text can be modified to make the delivery more direct, the manuscript will be much improved.

Reviewer #3:

Remarks to the Author:

This study investigates a crucial problem of the anti-angiogenic therapy that is the development of resistance to treatment in endothelial cells.

This research beautifully analyses the molecular mechanisms of endothelial cell resistance to anti-angiogenic treatment identifying novel mediators and molecular interactions supporting such response.

The conceptual approach is original and the conclusions supported by focused and well-integrated multiple experimental approaches.

The manuscript would acquire further impact if some of the mechanisms identified in vitro could be validated in situ in sections of human tumors. The ideal experiment would be to compare human tissue from patients treated or not with anti-angiogenic therapies and expressing or not resistance to the treatment.

Original reviewers' comments:

Reviewer #1 (Remarks to the Author):

The manuscript by He et al. describes a highly novel study to investigate and identify mechanisms by which endothelial cells overcome growth inhibition mediated by the anti-angiogenic agent bevacizumab. This study uses a range of cutting edge technologies and approaches which are applicable to wide range of biological questions beyond this study of resistance to vascular endothelial growth factor (VEGF)-targeted anti-cancer therapy.

The authors have developed a system in which they are able to culture endothelial cells at high density allowing a kinome-wide CRISPR-Cas9 screen to be undertaken. Cells that preferentially grew in the presence of anti-VEGF antibodies were analysed to determine which kinases had been targeted, and candidate kinases were validated using siRNA mediated knockdown experiments. Some of these kinases included members of the bromodomain and Extra-Terminal motif (BET) protein family involved in chromatin regulation. The use pharmacological inhibitors to these proteins enabled confirmation of the role of these kinases in mediating resistance to anti-VEGF treatment. The paper explored how inhibiting these proteins affected endothelial proliferation, motility and cell cycle progression and sought to understand using next generation sequencing how inhibiting these BET proteins enabled endothelial cells to overcome VEGF blockade. This resulted in the identification of genes involved in modulating chromatin activity and these findings were again interrogated using small molecular inhibitors targeting class II histone deacetylases and deposition of new centromere proteins.

The greatest strength of this paper is its experimental approach to identify kinases involved in a clinically relevant biological question, and then to further use next generation sequencing to identify how they might be acting. Its weakness is that the findings have not been explored in any detail and this is acknowledged in the manuscript. This would involve considerably more work that is beyond the scope of this study. This study contributes to our understanding of our understanding of how endothelial cells may adapt to VEGF targeted therapy.

Overall this is an excellent piece of work and the wider applicability of the approaches to many areas of biology make this highly suitable for publication in Communications Biology.

Some more specific recommendations/ comments

In the results commentary the cells used are described as endothelial cells, some reference is made to immortalisation in the legend of figure 1 and indeed in the Materials and Methods section it notes that the cells used are an immortalised cell line derived originally from dermal microvascular cells. Typically, studies with endothelial cells use primary cells and so it would be good to be clearer on this point particularly in the main results commentary. While it understandable that an immortalised cell line would be necessary for the CRISPR-Cas9 screen, the other studies, such as the siRNA transfection confirmatory experiments, should ideally be done with primary endothelial cells (to ensure that the findings from the first screen were not related to

endothelial cell transformation). Clarification of the cells used throughout would be useful and if the immortalised cell line was used throughout, this should be acknowledged and the associated limitations discussed.

This is a complex study and the authors have included a number of useful schematics which aid the reader in understanding the various experimental protocols. The reader would be additionally helped if the figure legends were more informative and summarised the findings demonstrated in the figure, and if all subsections of the results had a summary at the end clearly explaining the findings of that section (this is there for many but not all sections). A flow chart of the entire study would also be useful.

As described in the methods sections the statistical approaches are sound, however they are not always applied in the way outlined in this section. For example, in figure 3d which contains multiple groups, t-tests rather than ANOVA were performed. In this particular study this is an additional problem in that the ratios generated with the different knockdowns are compared to 1, however this does not account for any variation that one would expect to see in a control sample where both sets of green and red fluorescent protein expressing cells are transfected with control siRNA duplex. Such a control for this experiment is not evident from the data, and it is important both for the statistical analyses and for ruling out the theoretical possibility that expression of different fluorescent proteins differentially affects the cell proliferation.

Reviewer #2 (Remarks to the Author):

In this manuscript, the authors describe an experimental analysis aimed at uncovering the mechanisms of resistance to anti-angiogenesis therapy. More specifically, they have assessed the pathways that underly resistance to the anti-VEGFA antibody bevacizumab and have done so using a CRISPR-Cas9 loss-of-function screen of the kinome and in vitro, 3D microcarrier-based culture of human microvascular endothelial cells. The major claim is that epigenetic reader proteins of the BET family can in part explain how MVECs respond to bevacizumab treatment.

Comments:

1. This analysis is one of very few to make progress in defining the mechanisms that underly resistance to anti-angiogenic treatment. This is a thorny issue because it is complex and because there are real-world implications for cancer treatment. This subject matter is thus of considerable importance and appropriate for a high-level journal.

2. This is an impressive study because the analysis is innovative, meticulous and by-and-large, thorough. Though this is a study based on culture analysis, it is well thought out and establishes a launching point for future in vivo experimentation that can assess the role of BET protein activities in resistance to anti-angiogenic therapy.

3. In Figure 3f, a no-drug control is missing. Where you have a nice dose response (as with compound 43) it may be a little academic, but with one of the compounds (JQ1) used throughout the rest of the analysis, there is no dose response, and so a no-drug control is quite important. This should be included for all f panels of this figure.

4. Please define the culture conditions for the experiments described in Figure 4. Obviously, the reader is anticipating an analysis of VEGFA-dependent culture conditions, but it isn't clear whether this is the case for the data in this figure.
5. The analysis described in Figure 5 appears to be valuable but is complex and it isn't always clear exactly what was done. My understanding is that all the data in panels b-e of Figure 5 were generated using the cell death assay described in Fig. 3e. If this is correct, please make sure this is explicitly stated. If this isn't correct, then I have nicely illustrated the problem for you.
6. Is there an assay for the BET proteins that more directly assesses their activity? As it stands, we don't have confirmation of this in these assays and we have to make the assumption that the inhibitors perform as advertised. Can you use a population of MVECs that are CRISPR targeted for BET gene loss-of-function as an adjunct to the pharmacological assays? The inhibitor assays are valuable in their own right, but if they were bolstered by genetic loss of functions (assays with similar responses) this would add a level of certainty to the interpretation.
7. In this sentence, "Noticeably, deposition of new CENP-A-containing nucleosomes.....", The first word should be "Notably, .."
8. Finally, the way this manuscript is written tends to make it somewhat opaque. There are many abbreviations and quite a bit of convoluted sentence structure. If the text can be modified to make the delivery more direct, the manuscript will be much improved.

Reviewer #3 (Remarks to the Author):

This study investigates a crucial problem of the anti-angiogenic therapy that is the development of resistance to treatment in endothelial cells. This research beautifully analyses the molecular mechanisms of endothelial cell resistance to anti-angiogenic treatment identifying novel mediators and molecular interactions supporting such response. The conceptual approach is original and the conclusions supported by focused and well-integrated multiple experimental approaches. The manuscript would acquire further impact if some of the mechanisms identified in vitro could be validated in situ in sections of human tumors. The ideal experiment would be to compare human tissue from patients treated or not with anti-angiogenic therapies and expressing or not resistance to the treatment.

Authors' response to reviewers' comments:

We would like to express our great gratitude to all three expert reviewers for their positive and encouraging comments on our work. We also highly appreciate the constructive questioning and insightful suggestions. In the following section, we provide our point-by-point responses to all reviewers' comments which are presented in the text box.

Reviewer #1 (Remarks to the Author):

The manuscript by He et al. describes a highly novel study to investigate and identify mechanisms by which endothelial cells overcome growth inhibition mediated by the anti-angiogenic agent bevacizumab. This study uses a range of cutting-edge technologies and approaches which are applicable to wide range of biological questions beyond this study of resistance to vascular endothelial growth factor (VEGF)-targeted anti-cancer therapy.

The authors have developed a system in which they are able to culture endothelial cells at high density allowing a kinome-wide CRISPR-Cas9 screen to be undertaken. Cells that preferentially grew in the presence of anti-VEGF antibodies were analysed to determine which kinases had been targeted, and candidate kinases were validated using siRNA mediated knockdown experiments. Some of these kinases included members of the bromodomain and Extra-Terminal motif (BET) protein family involved in chromatin regulation. The use pharmacological inhibitors to these proteins enabled confirmation of the role of these kinases in mediating resistance to anti-VEGF treatment. The paper explored how inhibiting these proteins affected endothelial proliferation, motility and cell cycle progression and sought to understand using next generation sequencing how inhibiting these BET proteins enabled endothelial cells to overcome VEGF blockade. This resulted in the identification of genes involved in modulating chromatin activity and these findings were again interrogated using small molecular inhibitors targeting class II histone deacetylases and deposition of new centromere proteins.

The greatest strength of this paper is its experimental approach to identify kinases involved in a clinically relevant biological question, and then to further use next generation sequencing to identify how they might be acting. Its weakness is that the findings have not been explored in any detail and this is acknowledged in the manuscript. This would involve considerably more work that is beyond the scope of this study. This study contributes to our understanding of our understanding of how endothelial cells may adapt to VEGF targeted therapy.

Overall this is an excellent piece of work and the wider applicability of the approaches to many areas of biology make this highly suitable for publication in Communications Biology.

Some more specific recommendations/comments:

1. In the results commentary the cells used are described as endothelial cells, some reference is made to immortalisation in the legend of figure 1 and indeed in the

Materials and Methods section it notes that the cells used are an immortalised cell line derived originally from dermal microvascular cells. Typically, studies with endothelial cells use primary cells and so it would be good to be clearer on this point particularly in the main results commentary. While it understandable that an immortalised cell line would be necessary for the CRISPR-Cas9 screen, the other studies, such as the siRNA transfection confirmatory experiments, should ideally be done with primary endothelial cells (to ensure that the findings from the first screen were not related to endothelial cell transformation). Clarification of the cells used throughout would be useful and if the immortalised cell line was used throughout, this should be acknowledged and the associated limitations discussed.

We thank the reviewer for pointing this out and understanding the necessity of using an immortalised cell line for the CRISPR–Cas9 screen. The human dermal microvascular blood endothelial cells (ECs) we used were licenced from Lonza (Cat. No. XSEB113C1) and were immortalised with hTERT which provides chromosomal stability over many cell divisions while maintaining the in vivo nature of primary cells. In addition, this cell line was validated by Lonza for important phenotypic features of primary blood ECs. For example, XSEB113C1 cells express the major pan-EC markers (e.g. VWF, VEGFR2, VE-cadherin, PECAM1) but not those considered lymphatic EC specific (e.g. prospero homeobox 1 (PROX1)), are responsive to VEGFA and can form tubes as well as has a finite life span in vitro (also see the table attached below). Indeed, we have shown that XSEB113C1 cells responded expectedly to bevacizumab in our VEGFA-dependent system (Results, Fig. 5). Furthermore, the studies mentioned in Discussion (Huang et al. 2016; Bid et al. 2016; Mumby et al. 2017) showed that BET inhibition using JQ1 suppressed proliferation, migration and cell cycle progression in different types of primary blood ECs (e.g. human umbilical vein vascular ECs and human pulmonary microvascular ECs). In agreement with these findings, our data also showed the same effects of BET inhibition on these key EC activities (Results, Fig. 4). These indicate the biological relevance of the cell line we used.

For a higher level of scientific rigour, we have included clear statements for the use of the immortalised EC line and discussed the limitations accordingly (Page 5, lines 110–112; Page 15, lines 413–417; Page 16, lines 443–445; Page 32, line 894).

III. Quality Control

Cat. No.	Description	Cells/Vial	Viability	Maximum Productive Population Doublings	Average Doubling Time	Properties
XSEB113C1	Dermal Blood Microvascular Endothelial (BEC)	≥500,000 cells	≥70%	40 PD	40 hrs	Ac LDL+, FVIII+, CD31+, CD34-, Podo-, VEGF responsive, form tubes.
XSEL6C1	Dermal Lymphatic Microvascular Endothelial (LEC)	≥500,000 cells	≥70%	60 PD	30 hrs	Ac LDL+, FVIII+, CD31+, CD34+, Podo+, VEGF responsive, form tubes.

All cells are performance assayed and test negative for HIV-1, mycoplasma, Hepatitis-B, Hepatitis-C, bacteria, yeast and fungi. Cell viability, morphology, cell number, and proliferative capacity are measured after recovery from cryopreservation. Clonetics™ Media are formulated for optimal growth of specific types of human cells. COAs for all media products are available upon request. Please see Section XII (Product Warranty, Page 6) for more information on Quality Control claims and guarantees.

(source: https://bioscience.lonza.com/lonza_bs/CA/en/document/27640)

2. This is a complex study and the authors have included a number of useful schematics which aid the reader in understanding the various experimental protocols. The reader would be additionally helped if the figure legends were more informative and summarised the findings demonstrated in the figure, and if all subsections of the

results had a summary at the end clearly explaining the findings of that section (this is there for many but not all sections). A flow chart of the entire study would also be useful.

We thank the reviewer for the helpful suggestions. We have improved the figure legends and ensured to include summary statements for each subsection of Results (Page 11, lines 313–315; Page 32; Page 35; Page 38; Page 41). We also added a flow chart of the entire study (as Supplementary Figure 1) to guide readers about the experimental procedures and the main findings (Page 4, lines 101–103).

Supplementary Figure 1. Flow chart of the study. In this study, we aim to investigate how human microvascular blood endothelial cells (ECs) response to anti-angiogenic therapy. To this end, we have established three-dimensional (3D) culture system to allow large-scale in vitro EC culture to meet the high demand for cell number in a CRISPR screen. We have also developed effective CRISPR–Cas9 gene editing in ECs and VEGFA-dependent culture conditions for assessing EC response to the VEGFA neutralizing antibody bevacizumab. We performed the 3D CRISPR screen by combining these three key elements and identified novel EC response modifiers to bevacizumab. Further candidate validation and characterization experiments confirmed interaction between bromodomain and extraterminal domain (BET) inhibition and VEGFA blockade which was associated with altered EC response. Mechanistically, this may be related to chromosomal regulation via CDC25B. Our study not only provides technical advances to enable large-scale in vitro screening in ECs but also reveals molecular insight into EC response to anti-angiogenic therapy. hTERT, human telomerase reverse transcriptase; MAGECK, Model-based Analysis of Genome-wide CRISPR/Cas9 Knockout; NGS, next-generation sequencing.

3. As described in the methods sections the statistical approaches are sound, however they are not always applied in the way outlined in this section. For example, in figure 3d which contains multiple groups, t-tests rather than ANOVA were performed. In this particular study this is an additional problem in that the ratios generated with the different knockdowns are compared to 1, however this does not account for any variation that one would expect to see in a control sample where both sets of green and red fluorescent protein expressing cells are transfected with control siRNA duplex. Such a control for this experiment is not evident from the data, and it is important both for the statistical analyses and for ruling out the theoretical possibility that expression of different fluorescent proteins differentially affects the cell proliferation.

We thank the reviewer for the positive comments and the specific concern about the endothelial cell-multicolour competition assay (EC-MCA).

For the results in Fig. 3d, the one EC-MCA ratio (M) generated by each siRNA pool was compared with 1, which is essentially a two-group comparison between two ratios. Such comparison for each siRNA pool was independent from each other (as each ratio was generated in independent experiments and only compared with 1). Hence, we performed t -test for this assay and presented results in one figure panel for readability (these results essentially can be presented in different figure panels). In contrast, we performed one-way ANOVA in Fig. 3f because the small-molecule inhibitor assay (SMIA) ratios generated by different doses of one drug were compared with 1. The ratios generated by these doses were related and were from the same experiment. Hence, this was considered as multiple comparisons. So if we used different concentrations of siRNA pool in the EC-MCA, we would perform one-way ANOVA to compare the ratios generated by different concentrations of the same siRNA pool.

One of the important features of the EC-MCA is that it is internally controlled, which means that the experimental and control cells (i.e. green cells transfected with experimental siRNAs and red cells with control siRNAs) are mixed together and therefore under the same conditions throughout the assay duration. In addition, the way that the final EC-MCA ratio is calculated takes the effects of fluorescence protein and the control antibody treatment (palivizumab) into account and makes sure they are well controlled. These features enable very robust assay outputs that only changes caused by experimental siRNAs specific to bevacizumab treatment would be detected. In fact, there were many ratios highly closed to 1 in Fig. 3d and this actually confirmed the robustness of the EC-MCA.

Reviewer #2 (Remarks to the Author):

In this manuscript, the authors describe an experimental analysis aimed at uncovering the mechanisms of resistance to anti-angiogenesis therapy. More specifically, they have assessed the pathways that underly resistance to the anti-VEGFA antibody bevacizumab and have done so using a CRISPR-Cas9 loss-of-function screen of the kinome and in vitro, 3D microcarrier-based culture of human microvascular endothelial cells. The major claim is that epigenetic reader proteins of the BET family can in part explain how MVECs respond to bevacizumab treatment.

Comments:

1. This analysis is one of very few to make progress in defining the mechanisms that underly resistance to anti-angiogenic treatment. This is a thorny issue because it is complex and because there are real-world implications for cancer treatment. This subject matter is thus of considerable importance and appropriate for a high-level journal.
2. This is an impressive study because the analysis is innovative, meticulous and by-and-large, thorough. Though this is a study based on culture analysis, it is well thought out and establishes a launching point for future in vivo experimentation that can assess the role of BET protein activities in resistance to anti-angiogenic therapy.

We thank the reviewer for the positive comments and the appreciation of the importance and novelty of our work. The main goal of our work is to take initiatives to understand the role of ECs in the tumour/disease response to anti-angiogenic therapy by establishing a large-scale screening platform in ECs. This was very challenging because of the difficulties in optimising large-scale in vitro culture of ECs as well as in defining the VEGFA-dependent culture conditions. In this study, we hope to provide a starting point for further pre-clinical study and clinical investigation by providing both technical advances and potential candidates for assessing drug resistance and developing biomarkers. Meanwhile, this work may provide useful information for other research areas to study complex biological questions and apply to solve, for example, real-world clinical problems.

3. In Figure 3f, a no-drug control is missing. Where you have a nice dose response (as with compound 43) it may be a little academic, but with one of the compounds (JQ1) used throughout the rest of the analysis, there is no dose response, and so a no-drug control is quite important. This should be included for all f panels of this figure.

We thank the reviewer for this comment. As shown in Fig. 3e, the small-molecule inhibitor assay (SMIA) includes vehicle (i.e. no-drug control), drug, control antibody treatment (palivizumab) and bevacizumab treatment. The calculation of the SMIA ratio (S) takes the effects of vehicle (i.e. no-drug control) and the control antibody treatment (palivizumab) into account. Any noise caused by these factors would be well controlled and cancelled out. Hence, the final ratio only reflects the changes made by the experimental drug that are specific to bevacizumab treatment. When treatment was vehicle (i.e. no-drug control), the final ratio was 1. To further clarify this, we have added bars for no-drug control in the updated figure (Fig. 3f).

Fig. 3 Validation of screen candidate genes.

4. Please define the culture conditions for the experiments described in Figure 4. Obviously, the reader is anticipating an analysis of VEGFA-dependent culture conditions, but it isn't clear whether this is the case for the data in this figure.

We thank the reviewer for this suggestion and we regret that this caused confusions. Fig. 4 is describing the results obtained under VEGFA-independent culture conditions (i.e. the EC complete growth medium, EGM-2MV) because purely the effects of targeting BET proteins were evaluated. The effects of BET inhibition under VEGFA-dependent conditions were shown in Fig. 5. We have added specific statements in both Results (Page 7, lines 207–208) and figure legend of Fig. 4 (Page 36, line 937) to make this unambiguous.

5. The analysis described in Figure 5 appears to be valuable but is complex and it isn't always clear exactly what was done. My understanding is that all the data in panels b-e of Figure 5 were generated using the cell death assay described in Fig. 3e. If this is correct, please make sure this is explicitly stated. If this isn't correct, then I have nicely illustrated the problem for you.

We thank the reviewer for this suggestion and we regret that this caused confusions. The data in Fig. 5 b–e were indeed generated using the SMIA described in Fig. 3e and the only difference was that multiple ratios were obtained on days 1, 3, 5, 6, 7 and 9 for each condition. We have added specific statements in Results (Page 8, line 229) to make this unambiguous.

6. Is there an assay for the BET proteins that more directly assesses their activity? As it stands, we don't have confirmation of this in these assays and we have to make the assumption that the inhibitors perform as advertised. Can you use a population of MVECs that are CRISPR targeted for BET gene loss-of-function as an adjunct to the pharmacological assays? The inhibitor assays are valuable in their own right, but if they were bolstered by genetic loss of functions (assays with similar responses) this would add a level of certainty to the interpretation.

We thank the reviewer for this comment. For validation of the screen results, we have performed EC-MCA which incorporated siRNA pools targeting *BRD2*, *BRD3* or *BRD4* individually. These results were in agreement with those from the inhibitor assay using JQ1 or I-BET762, confirming the findings from different perspectives. The similar EC-MCA ratios from targeting *BRD2*, *BRD3* and *BRD4* individually suggest a common role of BET proteins in affecting the EC response to bevacizumab. We therefore used pan-BET inhibitors JQ1 and I-BET762 in the subsequent investigation such as RNA-Seq. However, it may gain further insight if effects of individual BRD protein (*BRD2*, *BRD3* or *BRD4*) and/or individual tandem bromodomain (BD1 or BD2, as discussed in Discussion) can be assessed. This may be a future experiment.

7. In this sentence, “Noticeably, deposition of new CENP-A-containing nucleosomes....., The first word should be “Notably, ..”

We have changed this word (Page 11, line 320).

8. Finally, the way this manuscript is written tends to make it somewhat opaque. There are many abbreviations and quite a bit of convoluted sentence structure. If the text can be modified to make the delivery more direct, the manuscript will be much improved.

We thank the reviewer for this suggestion and we have made several modifications (Page 2, lines 36, 37 and 39; Page 3, lines 56–57 and 65; Page 4, lines 85 and 94; Page 5, lines 115, 123, 125, 126 and 131–133; Page 10, lines 292 and 294; Page 11, line 318; Page 14, line 401).

Reviewer #3 (Remarks to the Author):

This study investigates a crucial problem of the anti-angiogenic therapy that is the development of resistance to treatment in endothelial cells. This research beautifully analyses the molecular mechanisms of endothelial cell resistance to anti-angiogenic treatment identifying novel mediators and molecular interactions supporting such response. The conceptual approach is original and the conclusions supported by focused and well-integrated multiple experimental approaches. The manuscript would acquire further impact if some of the mechanisms identified in vitro could be validated in situ in sections of human tumors. The ideal experiment would be to compare human tissue from patients treated or not with anti-angiogenic therapies and expressing or not resistance to the treatment.

We thank the reviewer for the positive comments and the appreciation of the importance and novelty of our work. For investigating EC response, in tumours for example, it may be difficult to gain useful information from previous and current clinical tumour databases which were usually established based on bulk tumour samples. However, with the development of single-cell sequencing technology, newly established databases would provide specific information regarding vasculature and ECs associated with tumours or other diseases treated with anti-angiogenic therapy. In the near-term, it would become more feasible to evaluate pre-clinical findings of EC signalling and activity in the clinical setting. This has been also discussed in Discussion.

Reviewers' Comments:

Reviewer #1:

Remarks to the Author:

I would like to thank the authors for addressing the points raised and I am happy with the responses and modifications to the manuscript.

Reviewer #2:

Remarks to the Author:

I am happy with the changes the authors have made in the revision.

Reviewer #3:

Remarks to the Author:

The authors have reasonably addressed the issue raised by the previous version of their manuscript.